# Astrocyte GluN2C NMDA receptors control basal synaptic strengths of hippocampal CA1 pyramidal neurons in the *stratum radiatum*

Peter H Chipman[1‡], Chi Chung Alan Fung[2†], Alejandra Pazo Fernandez[1†], Abhilash Sawant[1§], Angelo Tedoldi[1], Atsushi Kawai[1], Sunita Ghimire Gautam[1#], Mizuki Kurosawa[1], Manabu Abe[3], Kenji Sakimura[3], Tomoki Fukai[2], Yukiko Goda[1]*

[1]RIKEN Center for Brain Science, Wako-shi, Saitama, Japan; [2]Neural Coding and Brain Computing Unit, Okinawa Institute of Science and Technology Graduate University, Onna-son, Japan; [3]Department of Animal Model Development, Brain Research Institute, Niigata University, Niigata, Japan

**\*For correspondence:**
yukiko.goda@riken.jp

[†]These authors contributed equally to this work

**Present address:** [‡]Department of Biochemistry and Biophysics, University of California, San Francisco, San Francisco, United States; [§]Department of Neuroscience, University of Wisconsin-Madison, Madison, United States; [#]Shikhar Biotech Pvt. Ltd, Lalitpur, Nepal

**Abstract** Experience-dependent plasticity is a key feature of brain synapses for which neuronal N-Methyl-D-Aspartate receptors (NMDARs) play a major role, from developmental circuit refinement to learning and memory. Astrocytes also express NMDARs, although their exact function has remained controversial. Here, we identify in mouse hippocampus, a circuit function for GluN2C NMDAR, a subtype highly expressed in astrocytes, in layer-specific tuning of synaptic strengths in CA1 pyramidal neurons. Interfering with astrocyte NMDAR or GluN2C NMDAR activity reduces the range of presynaptic strength distribution specifically in the *stratum radiatum* inputs without an appreciable change in the mean presynaptic strength. Mathematical modeling shows that narrowing of the width of presynaptic release probability distribution compromises the expression of long-term synaptic plasticity. Our findings suggest a novel feedback signaling system that uses astrocyte GluN2C NMDARs to adjust basal synaptic weight distribution of Schaffer collateral inputs, which in turn impacts computations performed by the CA1 pyramidal neuron.

## Editor's evaluation

This paper provides evidence that NMDA receptors containing the GluN2C subunit are expressed in hippocampal astrocytes and are involved in maintaining a wide distribution of presynaptic release probabilities at local excitatory synapses. Theoretical modelling suggests this wide distribution of synaptic efficacy is conducive to the expression of the synaptic plasticity. It will be interesting to see if future studies support a role for astrocytic NMDA receptors in memory storage.

## Introduction

Plasticity is a fundamental feature of neuronal connections in the brain, where experience-dependent changes in synaptic strengths over different time scales are crucial for a variety of processes ranging from neural circuit development, circuit computations to learning and memory (*Feldman and Brecht, 2005*; *Bliss and Collingridge, 1993*; *Abbott and Regehr, 2004*; *Collingridge et al., 2010*; *Nicoll, 2017*). Deciphering how neurons dynamically express different forms of synaptic plasticity while ensuring the optimal performance of the circuit remains a key challenge (*Vitureira and Goda, 2013*; *Turrigiano, 2017*; *Nicoll, 2017*; *Brunel, 2016*). In particular, the distribution of synaptic strengths is

thought to reflect the capacity or the state of information storage of neural circuits (*Barbour et al., 2007*; *Buzsáki and Mizuseki, 2014*; *Bromer et al., 2018*). A better understanding of the cellular mechanisms that regulate synaptic strength distribution could therefore provide novel insights into the basis for the effective execution of circuit operations (*Barbour et al., 2007*).

The N-methyl-D-aspartate receptors (NMDARs) are a major ionotropic glutamate receptor type mediating excitatory synaptic transmission (*Paoletti et al., 2013*; *Sanz-Clemente et al., 2013*; *Hansen et al., 2018*). NMDARs are heteromeric assemblies consisting of four subunits, and seven NMDAR subunit genes have been identified that fall into three subfamilies: *Grin1* encoding the obligatory GluN1 subunit, four *Grin2* genes encoding GluN2A-D, and two *Grin3* genes encoding GluN3A-B (*Paoletti et al., 2013*; *Hansen et al., 2018*). The differing subunit compositions confer NMDARs with distinct biophysical and pharmacological properties and contribute to their diverse biological activities. NMDARs are expressed throughout the central nervous system (CNS) and are crucial for normal brain function and plasticity. In particular, NMDARs that are typically composed of two GluN1 and two GluN2 subunits and are present postsynaptically, have been extensively studied for their role in memory mechanisms (*Paoletti et al., 2013*; *Sanz-Clemente et al., 2013*; *Nicoll, 2017*). NMDARs are also involved in pathological conditions such as epilepsy, ischemia, and traumatic brain injury (*Hansen et al., 2017*). In stark contrast to the multifaceted functions of neuronal NMDARs, very little is known of NMDAR functions in glial cells in the CNS. Amongst the better characterized glial NMDARs, NMDARs in oligodendrocytes play a role in axonal energy metabolism (*Saab et al., 2016*) and exacerbate pathological conditions (*Káradóttir et al., 2005*). In astrocytes, NMDARs have been long thought to be absent except following ischemia in vivo or under anoxia in vitro that promote their aberrant expression (*Krebs et al., 2003*; *Gottlieb and Matute, 1997*). Nevertheless, there are reports of GluN2C mRNA expression in astrocytes in the adult brain (*Watanabe et al., 1993*; *Karavanova et al., 2007*; *Ravikrishnan et al., 2018*), and several studies provide physiological evidence in support of astrocyte NMDAR expression under non-pathological conditions (*Schipke et al., 2001*; *Serrano et al., 2008*; *Lalo et al., 2006*; *Letellier et al., 2016*). The precise function for astrocyte NMDARs, however, has remained a matter of debate (e.g. *Kirchhoff, 2017*), and the role for astrocyte GluN2C has not yet been identified.

Astrocytes regulate a diverse set of essential brain activities, from neurovascular coupling, metabolic support, and ionic homeostasis (e.g. *Attwell et al., 2010*; *Giaume et al., 2010*; *Simard and Nedergaard, 2004*) to synaptic connectivity and function (e.g. *Araque et al., 2014*; *Clarke and Barres, 2013*). Recent studies have highlighted a role for astrocytes in the bidirectional control of basal synaptic transmission (*Panatier et al., 2011*; *Perea and Araque, 2007*; *Di Castro et al., 2011*; *Martin-Fernandez et al., 2017*; *Schwarz et al., 2017*), which likely involves signaling via their perisynaptic processes (*Bindocci et al., 2017*; *Bazargani and Attwell, 2016*). Moreover, mounting evidence suggests that astrocytes are not only capable of either simply potentiating (*Panatier et al., 2011*; *Perea and Araque, 2007*; *Jourdain et al., 2007*) or depressing synaptic transmission (*Martín et al., 2015*; *Pascual et al., 2005*), but a single astrocyte, which contacts tens of thousands of synapses, can concurrently mediate bi-directional synapse regulation at separate synaptic contact sites (*Schwarz et al., 2017*; *Covelo and Araque, 2018*). Astrocytes likely regulate neural circuit functions at multiple levels, as suggested by observations of both local and global astrocyte signaling that are triggered in a synapse, neuron, or circuit-specific manner (*Martin-Fernandez et al., 2017*; *Martín et al., 2015*; *Chai et al., 2017*; *Deemyad et al., 2018*; *Dallérac et al., 2018*; *Santello et al., 2019*). However, despite these advances, the basic mechanisms by which astrocytes detect and adjust synaptic transmission and the extent of their impact on local synaptic circuit activity are not fully understood.

Astrocytes express a plethora of neurotransmitter receptors and membrane transporters that are thought to modulate synaptic transmission (*Araque et al., 2014*; *Bazargani and Attwell, 2016*), including perisynaptic astrocyte glutamate transporters that influence the magnitude and kinetics of postsynaptic glutamate receptor activation and membrane depolarization (*Pannasch et al., 2014*; *Murphy-Royal et al., 2015*) and astrocyte metabotropic glutamate receptors (mGluRs) whose activation promote gliotransmitter release to provide feedback control of synaptic transmission (*Panatier et al., 2011*; *Schwarz et al., 2017*; *Covelo and Araque, 2018*; *Araque et al., 2014*; *Bazargani and Attwell, 2016*). Given the often shared expression of neurotransmitter receptors and transporters between astrocytes and neurons, an important challenge is to decipher the synaptic activity-dependent functions of astrocyte receptors and transporters independently of the functions of their

neuronal counterparts. Notably, many of the prior studies have focused on pathological brain states (*Krebs et al., 2003*; *Gottlieb and Matute, 1997*) or used young brain tissue (e.g. *Panatier et al., 2011*) or culture preparations (e.g. *Schwarz et al., 2017*) in which the expression pattern of the astrocyte receptors and channels differ substantially from the adult brain in basal conditions (*Sun et al., 2013*; also see *Boisvert et al., 2018*). Consequently, the cellular basis and the network consequences of astrocyte-synapse interactions mediated by the astrocyte receptors and channels in the healthy adult brain remain to be clarified.

In a previous study, we identified a role for astrocyte signaling in regulating synaptic strength heterogeneity in hippocampal neurons, which involved astrocyte NMDARs (*Letellier et al., 2016*). Heterogeneity was assessed by comparing the presynaptic strengths of two independent inputs to pyramidal neurons, which was estimated by the paired-pulse ratio (PPR) of excitatory postsynaptic current (EPSC) amplitudes, a parameter inversely related to presynaptic release probability (*Dobrunz and Stevens, 1997*). Although basal PPRs were uncorrelated between the two inputs, surprisingly, inhibiting astrocyte NMDARs promoted the correlation of PPRs. This indicated that astrocyte NMDARs contributed to enhancing the differences in basal presynaptic strengths represented by the two different inputs (*Letellier et al., 2016*). Such pair-wise comparison of PPRs, however, is limited to providing an indirect measure of variability, and it remains unclear how astrocyte NMDARs control the overall shape of the PPR distribution. Furthermore, the crucial roles for neuronal NMDARs in regulating synaptic transmission and plasticity have confounded the interpretation of astrocyte NMDAR functions, which still remain enigmatic (*Kirchhoff, 2017*). Here, we further investigated astrocyte NMDAR-dependent regulation of synaptic strength in hippocampal CA1 pyramidal neurons in adult mice, focusing on the mode of regulation of presynaptic strength distribution, the NMDAR type responsible, and probing the potential layer specific regulation in area CA1. Our findings identify a role for astrocyte GluN2C NMDARs in maintaining broad presynaptic strength diversity specifically in the *stratum radiatum* (SR) by enhancing both strong and weak synapses. Mathematical modeling suggests that presynaptic strength diversity strongly influences the expression of long-term synaptic plasticity, which in turn, is important for network stability and learning and memory (*Malenka and Bear, 2004*; *Collingridge et al., 2010*; *Royer and Paré, 2003*; *Zenke and Gerstner, 2017*). Our findings suggest astrocyte GluN2C NMDARs as a key player in linking the regulation of synaptic strength distribution to the expression of synaptic plasticity that promotes optimal circuit performance.

## Results

### NMDAR-dependent regulation of PPR variability in hippocampal CA1 pyramidal neurons

In order to characterize the properties of astrocyte NMDAR-dependent modulation of synaptic strength, here we devised a simplified assay using a single stimulating electrode to monitor PPR variability of Schaffer collateral synapses. Hippocampal slices were prepared from adult mouse brain (P60-120), and EPSCs to a pairwise stimulation (50 ms inter-stimulus interval) of Schaffer collateral axons were recorded in CA1 pyramidal neurons (*Figure 1A*; *Figure 1—figure supplement 1*). MK801 (1 mM) was infused into the postsynaptic cell via the patch pipette, and after 10–15 min but prior to starting the experiment, Schaffer collaterals were stimulated at 0.1 Hz for at least 45 times to pre-block synaptic NMDARs. This resulted in over 92 % inhibition of synaptic NMDAR currents (*Bender et al., 2006*; *Figure 1—figure supplement 1*; see Materials and Methods). Upon confirming the block of postsynaptic NMDARs, the baseline EPSC recordings were taken; subsequently, NMDAR inhibitors were bath applied to test their effects on PPR variance. Under the block of postsynaptic NMDARs, any effects of NMDAR inhibitors, if observed, would be expected to reflect the influence of NMDARs present either in astrocytes or the presynaptic neurons. Notably, in mature brains which we used for our experiments, the contribution of presynaptic NMDARs in general had been suggested to be minimal (*Shih et al., 2013*). Upon bath applying MK801 (50 µM) or AP5 (50 µM), the population variance ($\delta^2$) of PPRs was reduced without a significant change in the mean ($\bar{x}$)(*Figure 1B–D*; MK801 $\delta^2$ $P$ = 0.016, $\bar{x}P$ = 0.856; AP5, $\delta^2$ $P$ = 0.047; $\bar{x}P$ = .162). This overall decrease in PPR variance, when examined at the level of individual inputs, was associated with a change in PPR (ΔPPR) that was negatively correlated to the baseline PPR: PPR of some inputs increased while others decreased by the NMDAR antagonist application (*Figure 1E–G*). Furthermore, consistent with a lack of change in the mean PPR

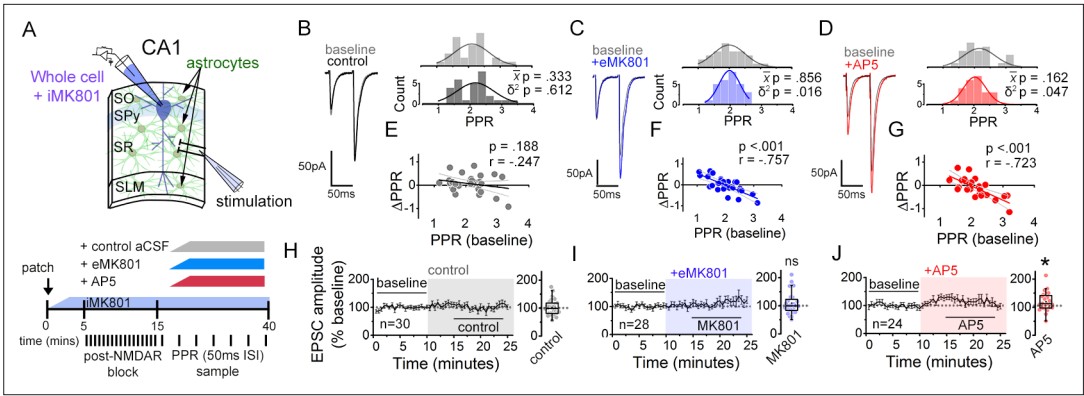

**Figure 1.** NMDARs contribute to synaptic strength diversity of Schaffer collateral synapses. (**A**) Experimental scheme to assess the role of non-postsynaptic NMDARs. MK801 was perfused through the patch pipette (iMK801) to the CA1 pyramidal neuron for 10–15 min, and pairs of stimuli (Δt = 50 ms) were applied at least 45 times at 0.1 Hz to pre-block postsynaptic NMDARs prior to bath applying NMDAR antagonists (bottom). (**B–D**) Left: representative EPSC traces (average of 20 sweeps) to pairs of pulses (Δt = 50 ms) applied to Schaffer collateral axons before (baseline) and after wash-on of vehicle control (**B**), external MK801 (eMK801: 50 μM) (**C**), or AP5 (50 μM) (**D**). Right: histograms of the PPRs recorded during the baseline and vehicle or drug application periods were fit with single Gaussian curves. The p values for the comparison of the baseline to the experimental periods for the population mean ($\bar{x}$) and variance ($\delta^2$) were obtained by two-tailed paired sample t-test and one-tailed f-test for equal variances, respectively. (**E–G**) Scatterplots of the change in PPR vs. the initial PPR in vehicle control (**E**), MK801 (**F**), or AP5 (**G**) where ΔPPR = PPR$_{experimental}$-PPR$_{baseline}$; linear fits, Pearson's correlation coefficients (**r**) and p values are as indicated. X intercept: MK801 = 1.97, AP5 = 1.95. (**H–J**) Plots of normalized EPSC amplitudes before and during the application of NMDAR antagonists (shaded area); n, number of inputs examined for each experiment. Baseline and experimental periods are indicated (black bars). Right: summary bar graph. * p < 0.05, Mann-Whitney U-test. Control = 30 inputs, 15 cells, 13 mice; MK801 = 28 inputs, 14 cells, 9 mice; AP5 = 24 inputs, 12 cells, 11 mice. Data are presented as mean ± s.e.m.

The online version of this article includes the following figure supplement(s) for figure 1:

**Figure supplement 1.** MK801 (1 mM) in the patch pipette (iMK801) completely blocks postsynaptic NMDARs.

**Figure supplement 2.** Antagonists of GluN2B-containing NMDARs (Ro25-6981) or mGluR5 (MPEP) do not alter synaptic weight diversity.

**Figure supplement 3.** NMDAR antagonists differentially influence presynaptic function.

**Figure supplement 4.** NMDAR antagonists have little effect on spontaneous neurotransmission and the waveform of evoked EPSCs.

**Figure supplement 5.** NMDAR antagonists that target GluN2C-containing NMDARs reduce PPR disparity in the *stratum radiatum* (SR).

---

upon blocking NMDARs, a linear fit to the data intercepted the x-axis near the baseline mean PPR (x-axis intercept: MK801 = 1.97; AP5 = 1.95).

We also tested for the contribution of GluN2B NMDAR and also of mGluR5 in maintaining PPR variance. Bath applying Ro25-6981 (10 μM), a GluN2B-specific NMDAR antagonist (*Fischer et al., 1997*), and MPEP (25 μM), an mGluR5-specific antagonist (*Gasparini et al., 1999*), had little effect on PPR variance (*Figure 1—figure supplement 2A-C*; Ro256981 p = 0.362, MPEP p = 0.349). MPEP, however, significantly increased the mean PPR (*Figure 1—figure supplement 2C*; p = 0.033), which is consistent with a reported role for mGluR5 in presynaptic strength regulation (*Panatier et al., 2011*).

We next examined whether the changes in PPR elicited by the extracellularly applied NMDAR inhibitors accompanied changes in the EPSC amplitude and the coefficient of variation (CV) of EPSC amplitude, the latter as an additional measure of the change in presynaptic release probability (*Malinow and Tsien, 1990*; *Larkman et al., 1992*). On average, bath applied MK801 did not substantially alter the amplitude of the first EPSC to the pair of stimuli relative to control slices, despite a small tendency for an increase with a time delay (*Figure 1I*; p = 0.657). MK801 also had little effect on CV$^{-2}$ of EPSC amplitudes (*Figure 1—figure supplement 3B*; p = 0.991), although the responses of individual inputs to MK801, albeit small in magnitude, were positively correlated to CV$^{-2}$ (*Figure 1—figure supplement 3D*; Pearson's *r* = 0.420, p = 0.026) and negatively correlated to PPR (*Figure 1—figure*

*supplement 3H*; Pearson's *r* = –0.474, p = 0.011). These data are consistent with the possibility that the observed reduction in PPR diversity by MK801 application is associated with normalization of presynaptic release probability.

In contrast to MK801, AP5 significantly increased EPSC amplitudes within 10 min of application (*Figure 1J*; p = 0.026), and the increase in $CV^{-2}$ (*Figure 1—figure supplement 3B*; AP5 p = 0.003) suggested that the change in EPSC amplitude involved a presynaptic change. Notably, Ro25-6981 that had no effect on PPR variance, did produce a significant increase in $CV^{-2}$ (*Figure 1—figure supplement 2H*; $CV^{-2}$ p = 0.002), and produced a small but non-significant increase in EPSC amplitudes (*Figure 1—figure supplement 2E*; EPSC p = 0.261), which was consistent with the observed increase in PPR (above). None of the NMDAR antagonists caused substantial changes in the amplitude or frequency of spontaneous EPSCs (sEPSCs) or evoked EPSC waveforms (*Figure 1—figure supplement 4*).

We next sought to confirm the NMDAR-dependent modulation of presynaptic strength diversity as described in our previous study by comparing PPRs across two independent Schafer collateral inputs that converge onto a target CA1 pyramidal neuron (*Letellier et al., 2016*), and to observe the time course of PPR population variance change by NMDAR antagonists. In this method, the absolute difference of PPRs between the two inputs (PPR disparity) is taken as a proxy for presynaptic strength variability, with lower PPR disparity indicating lower PPR variability and vice versa (*Figure 1—figure supplement 5A*,B). Note that graphs of the PPR difference contain a regression-to-the-mean component, which is however effectively cancelled out when making direct graph comparisons, within the same parametric space. Consistent with our analysis thus far, bath application of MK801 and AP5 reduced PPR disparity (*Figure 1—figure supplement 5D*,E; MK801 p = 0.001, AP5 p = 0.004), while Ro25-6981 and MPEP did not (*Figure 1—figure supplement 5G*,H; Ro25-6981 p = 0.670, MPEP p = 0.263).

Collectively, these findings support the idea that NMDARs play a role in regulating presynaptic strengths to broaden the variability of PPR without appreciably impacting the mean PPR. Moreover, such an involvement of NMDARs in presynaptic regulation is distinct from the presynaptic regulation by mGluR5 that potentiates synaptic strength across a synapse population.

## Astrocyte NMDARs mediate the effects of NMDAR antagonists on PPR diversity

To test the extent to which bath applied NMDAR antagonists affected synaptic transmission by targeting astrocyte NMDARs, we knocked down in astrocytes, the *Grin1* gene that encodes GluN1, a requisite subunit for the cell surface expression of NMDARs (*Fukaya et al., 2003*; *Abe et al., 2004*). AAV carrying either mCherry-tagged nlsCre (Cre) or a control nls-mCherry lacking Cre or in some experiments control EGFP lacking Cre (Control), all of which were expressed under the GFAP104 promoter, was injected bilaterally into the dorsal hippocampus of adult *Grin1* floxed mice (*Figure 2A*; see Materials and Methods section and E-phys statistics report for details)(*Letellier et al., 2016*). We used low titer AAVs (~0.4–4×$10^{10}$ genome copies/injection) to avoid potential reactive astrocytosis, and although the extent was modest, we obtained a highly specific mCherry expression in astrocytes throughout the area CA1 with little expression in neurons (*Figure 2B*: Cre, 41.1% ± 2.4% of GFAP+ ve cells [n = 3,529], 0.3% ± 0.1% of NeuN+ ve cells [n = 1,150] from 5 mice; Control, 40.7% ± 1.9% of GFAP+ ve cells [n = 3,309], 0.5% ± 0.1% of NeuN+ ve cells [n = 1,217] from 5 mice). The efficacy of GluN1 knock-down in astrocytes was assessed electrophysiologically by patch-clamping astrocytes and monitoring slow depolarizing responses elicited by puff applying NMDA and glycine (1 mM each) (*Figure 2—figure supplement 1*) which were mediated by NMDARs but were also contributed in part by voltage-gated calcium channels in astrocytes (*Letellier et al., 2016*). Astrocytes that expressed Cre as identified by the mCherry fluorescence in *stratum radiatum* (SR) where Schaffer collateral synapses were present, showed depolarizing responses to NMDA-glycine puff that were significantly decreased compared to controls (*Figure 2—figure supplement 1A*; P = 0.006). Given that mCherry-positive astrocytes were present throughout the CA1, we also monitored NMDA-glycine puff responses in astrocytes in *stratum oriens* (SO) and *stratum lacunosum moleculare* (SLM). In contrast to mCherry-positive astrocytes in SR, mCherry-labeled astrocytes in SO or SLM showed depolarizing responses to NMDA-glycine puff that were not different between Cre and Control slices (*Figure 2—figure supplement 1B, C*; SO p = 0.247; SLM p = 0.268), despite the fact that viral transduction efficiencies were

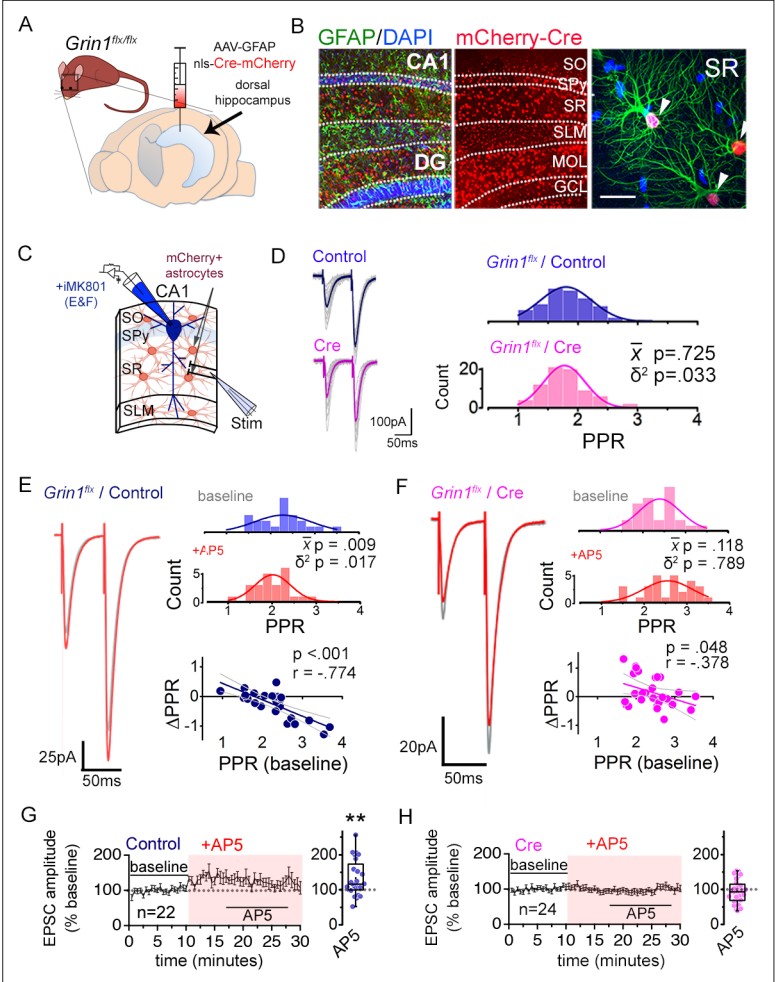

**Figure 2.** Astrocyte NMDARs mediate the antagonist-induced decrease in synaptic strength diversity. (**A**) Astrocyte GluN1 was conditionally knocked down by injecting AAV-GFAP104-nls-mCherry-Cre (Cre), or as a control, AAV-GFAP104-nls-mCherry (Control), to the dorsal hippocampus of adult $Grin1^{flx/flx}$ mice. (**B**) Representative image of hippocampal area CA1 immunolabeled for GFAP (green) and mCherry (red) and counterstained with DAPI (blue), 21 days after virus injection. SO, *stratum oriens*; SPy, *stratum pyramidale*; SR, *stratum radiatum*; SLM, *stratum lacunosum moleculare*; MOL, molecular layer; GCL, granule cell layer. Right: magnified view in SR showing restricted expression of mCherry to GFAP-positive astrocytes (arrowheads). Scale bar, 20 µm. (**C**) Recording scheme: CA1 pyramidal neurons were patched in acute slices prepared 21 days after Control or Cre AAV infection. EPSCs were elicited by placing the stimulating electrode in the SR. Patch pipette contained MK801 (iMK801) for experiments in *E-H*. (**D**) Left: Averaged EPSC traces from representative recordings in slices from Control (top) or Cre (bottom) virus-infected mice. Individual traces are shown in grey. Right: Histograms of PPRs recorded from control (top) and Cre slices (bottom) were fit with single Gaussians. The p values were obtained for the population mean ($\bar{x}$) (two-tailed t-test) and variance ($\delta^2$) (one-tailed f-test for equal variances) comparing PPR distributions from control (n = 72 inputs, 43 cells, 14 mice) and Cre (n = 74 inputs, 46 cells, 12 mice) slices. (**E,F**) Left: average EPSC traces in baseline (grey) and after drug treatment (color) from a representative experiment. Right: PPR histograms (top) and ΔPPR vs. baseline PPR plots (bottom) in baseline and after bath applying AP5 (50 µM) to control (**E**) and Cre (**F**) slices. The p values for the population mean ($\bar{x}$) and variance ($\delta^2$) are as in *D*, comparing PPRs before and after the drug treatment. Linear fits and Pearson's correlation coefficients (**r**) and p values are shown in scatter plots. (**G,H**) Plots of normalized EPSC amplitudes before and during the application of AP5 to control (**G**) and Cre-infected (**H**) slices (shaded area); n is the number of inputs examined. Right: summary bar graph. **p = 0.005, Mann-Whitney U-test. Control+ AP5 = 22 inputs, 12 cells, 5 mice; Cre+ AP5 = 24 inputs, 14 cells, 5 mice. Data are presented as mean ± s.e.m.

The online version of this article includes the following figure supplement(s) for figure 2:

**Figure supplement 1.** Puff application of NMDA and glycine triggers the depolarization of astrocytes in

*Figure 2 continued on next page*

*Figure 2 continued*

hippocampal CA1 in acute slices.

**Figure supplement 2.** Knock-down of GluN1 in hippocampal astrocytes selectively alters the PPR disparity without affecting the basic properties of synaptic transmission.

comparable amongst astrocytes across the three layers (SR: Cre 39.8% ± 2.4%, Control 40.2% ± 2.0%; SO: Cre 44.8% ± 3.0%, Control 39.3% ± 5.9%; SLM: Cre 39.1% ± 4.9%, Control 40.0% ± 4.0%). The input resistance of patched astrocytes was not altered by Cre expression in any layer (*Figure 2—figure supplement 1D*). Notably, NMDA-glycine responses in SLM were substantially smaller than those in SR or SO (*P* = 0.001, one way ANOVA, Bonferroni post-hoc tests). Nevertheless, AP5 bath application blocked NMDA-glycine puff-induced depolarizing responses in astrocytes in all three layers. This observation suggests that NMDARs are broadly expressed across hippocampal CA1 astrocytes. Moreover, the differential sensitivity of the slow depolarizing responses to astrocyte GluN1 knock-down between layers supports the possibility that the cumulative actions of signaling downstream to NMDAR activation including the contribution of voltage-gated conductances (*Serrano et al., 2008*; *Letellier et al., 2016*; *Shih et al., 2013*) could be heterogeneous across layers. We will return to this layer-specific differences later.

Having detected GluN1-dependent functional NMDAR activity in SR CA1 astrocytes, we next asked whether the GluN1 knock-down also affected synaptic transmission of Schaffer collateral synapses similarly to the bath applied NMDAR antagonists. Baseline PPR showed a significantly reduced PPR variance in Cre-infected slices compared to control slices without a change in the mean PPR (*Figure 2C and D*; $\delta^2$ *P* = 0.033; $\bar{x}P$ = 0.725). Analysis of the relative PPR from two independent Schaffer collateral inputs also revealed lower PPR disparity in Cre injected slices compared to control slices (*Figure 2—figure supplement 2A*; p = 0.009). These observations are consistent with a reduced PPR variance upon compromising astrocyte NMDAR signaling as observed with bath applied NMDAR inhibitors. The amplitude and frequency of spontaneous EPSCs and the waveform of evoked EPSCs were unaltered by astrocyte-specific GluN1 knock-down (*Figure 2—figure supplement 2B*,C).

We next determined whether astrocyte GluN1 knock-down could occlude the effects of bath applied AP5 on PPR variance, by recording from postsynaptic CA1 neurons infused with MK801 in Control and Cre-infected slices. Similar to naive slices, in Control slices, AP5 decreased the population variance of PPRs by strengthening some synapses and weakening others (*Figure 2E*; Control + AP5 p = 0.017). In addition, AP5 caused also a variable increase in EPSC amplitudes (*Figure 2G*; AP5 p = 0.005) that occurred without a concomitant change in the amplitude or the frequency of spontaneous EPSCs (*Figure 2—figure supplement 2D*,E). In Cre slices, however, the decrease in PPR variance by AP5 was strongly attenuated, despite the modest extent of virus infection in Cre slices (*Figure 2F*; Cre + AP5 p = 0.789). The analysis of PPR disparity across two inputs also showed substantially decreased inhibitory effect of AP5 in Cre slices relative to controls (*Figure 2—figure supplement 2H*,I; Control + AP5 p = 0.013, Cre + AP5 p = 0.315). Curiously, the small increase in EPSC amplitudes observed upon applying AP5 was also attenuated in Cre slices (*Figure 2H*). This suggests that although the effect of NMDAR antagonists on PPR diversity and EPSC amplitudes are likely to target distinct compartments – presynaptic and postsynaptic – both mechanisms may involve astrocyte NMDARs. Together, these observations indicate that astrocyte NMDARs are the major mediators of the synaptic effects of acutely blocking NMDARs in our experimental conditions, and further support the view that astrocyte NMDARs help maintain the broad variability of presynaptic strengths across a synapse population.

## Modelling of release probability variability reveals its impact on synaptic plasticity

In order to gain an insight into the physiological role for the broad release probability distribution, we sought to assess the impact of reducing the presynaptic strength variability on a synaptic learning rule by constructing a mathematical model. We first modelled release probability variations based on gamma distributions (*Buzsáki and Mizuseki, 2014*; *Branco and Staras, 2009*; *Murthy et al., 1997*). To achieve this, we experimentally obtained estimates of release probability using the styryl dye, FM1-43 in synaptic networks of hippocampal pyramidal neurons co-cultured with astrocytes. Briefly, neurons were extracellularly stimulated with 40 action potentials at 20 Hz in the presence of FM1-43 to label the

readily releasable vesicle pool, a parameter related to release probability (*Rosenmund and Stevens, 1996*; *Letellier et al., 2019*). Two sets of data were obtained: one in control condition containing 10 μM CNQX to prevent recurrent activity ('control'), and another in the additional presence of 50 μM AP5 ('AP5'). Because the block of synaptic AMPARs by CNQX would compromise the activation of neuronal NMDARs, we reasoned that any changes to the FM1-43 signal in the AP5 condition (i.e. CNQX + AP5) relative to the control could suggest the contribution of astrocyte NMDARs to the readily releasable pool size. The gamma distributions that best fit the FM1-43 signal distributions in control and AP5 groups were then obtained to model the release probability distributions in the simulations; the optimal parameters for the gamma distributions were determined by maximizing the likelihood function for each case (*Figure 3—figure supplement 1*; see Materials and methods). Notably, the gamma distribution for the AP5 condition showed a smaller variance than that of the control condition, which was reminiscent of the effect of bath applied AP5 on PPR distribution. To facilitate the cross-check of whether the reduction in variance per se was an important factor affecting the synaptic learning rule in the simulations, we also considered a gamma distribution that best fit the FM1-43 signal distribution in the AP5 condition by fixing the mean of the gamma distribution to remain the same as for the control condition ('AP5-with-control mean'; *Figure 3—figure supplement 1B*).

We next asked whether the influence of release probability variance on the efficacy of long-term synaptic plasticity could be observed in a computational model by simulating spike-timing dependent plasticity (STDP) using two leaky integrate-and-fire (LIF) neurons with a synaptic connection (*Figure 3A*, see Materials and methods). The presynaptic neuron received a spike train of magnitude $A_1$ = 500 pA, while the postsynaptic neuron received a spike train of variable magnitude $A_2$. Pairing 10 spikes at 20 Hz with a time difference between spike trains of $\Delta t$ resulted in LTP or LTD, with positive $\Delta t$ values producing LTP and negative $\Delta t$ values resulting in LTD (*Figure 3B*). Simulations using the decreased release probability variance of the AP5 condition showed little change in LTD whereas LTP was compromised compared to the control condition. Specifically, when the efficacy of synaptic weight change was compared as a function of the amplitude of the postsynaptic current injection $A_2$, a less variable release probability represented by the AP5 condition required higher levels of current injection to trigger LTP compared to a broad release probability distribution represented by the control condition (*Figure 3B*, left versus middle panel). The effect of narrowed release probability variance in increasing the threshold for eliciting LTP persisted also for the AP5-with-control mean group (*Figure 3B*, right panel). A further comparison was made by plotting the averages of the maximum change of synaptic weight for different $\Delta t$ when the $A_2$ was between 180 pA and 200 pA for the three conditions (*Figure 3C*). The analysis clearly revealed that the potentiation was more sensitive for smaller input current impulses. In effect, the results showed that the control condition exhibited a higher propensity to undergo LTP compared to the two AP5 groups representing conditions of reduced release probability variance.

We also explored the impact of narrowing the release probability variance under a more general STDP scenario by examining synaptic weight changes elicited in a pair of neurons receiving Poisson spike trains (*Figure 4A*). Specifically, the synaptic weights obtained in response to the same Poisson sequences in the control condition compared to the AP5 condition for independent simulations showed that the extent of potentiation was consistently smaller in the AP5 group (*Figure 4B*, left panel). Similarly, the extent of depression was also found decreased in the AP5 group (*Figure 4B*, left panel). These findings suggested that the reduced release probability variance could reshape the expression of long-term plasticity. Whether the reduced efficacy of plasticity expression could be attributed to the change in release probability variance per se was confirmed by testing also the AP5-with-control mean condition (*Figure 4B*, right panel), which showed attenuation of both LTP and LTD. When the absolute average difference in the extent of LTP or LTD relative to control was compared between the two AP5 conditions, reducing the variance of the release probability distribution without a change in the mean had a stronger effect in compromising LTP than LTD, while in the AP5 condition, both potentiation and depression were similarly affected (*Figure 4C*). Collectively, the results of the simulations suggest that a broader distribution of release probability could serve to promote the expression of long-term plasticity.

## Hippocampal astrocytes express GluN2C NMDARs

Our results thus far point to astrocyte NMDARs, and in particular, the involvement of the obligatory subunit GluN1 in regulating the variability of presynaptic strengths. In order to obtain direct molecular

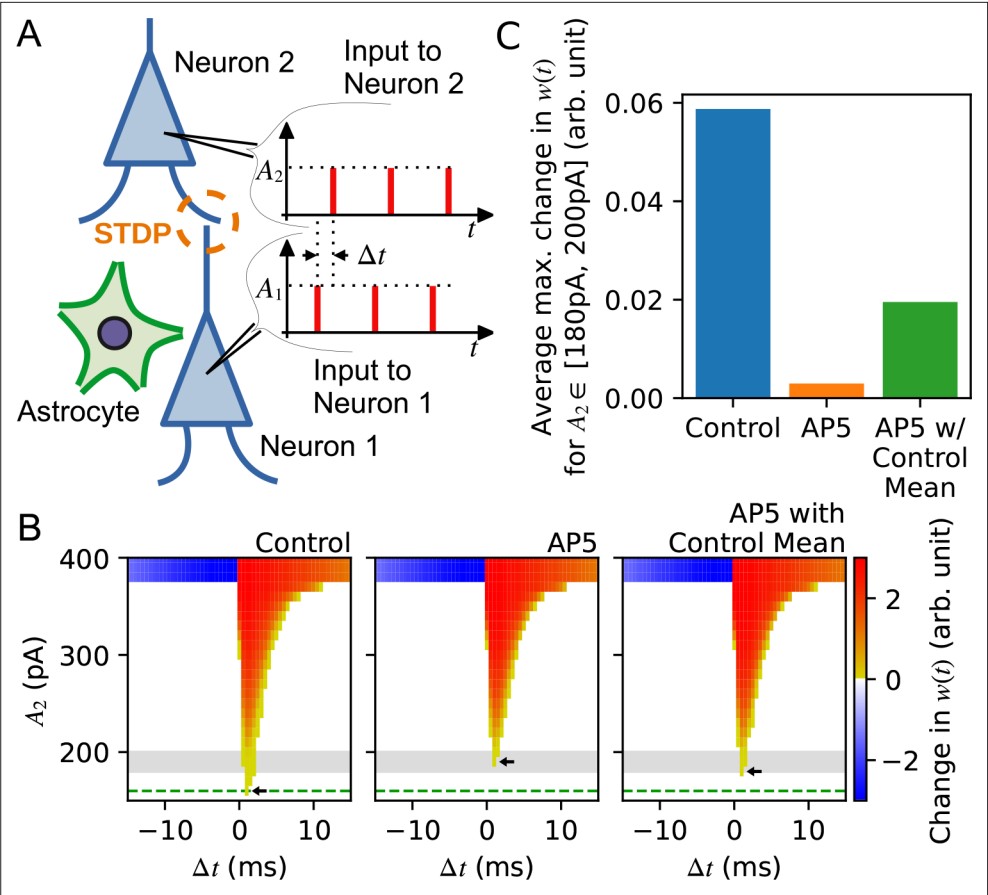

**Figure 3.** Synaptic neurotransmitter release probability diversity impacts long-term synaptic plasticity. (**A**) Illustration of the model used in the simulation. There are two leaky integrate-and-fire neurons with a synaptic connection. Each neuron receives a current injection whose magnitude is $A_1$ for neuron 1 and $A_2$ for neuron 2. The input spike trains for neuron 1 and neuron 2 are misaligned by $\Delta t$. $A_1$ is set at 500 pA which is large enough to trigger neuron 1 to spike, while $A_2$ is varied to test the sensitivity of STDP between neurons. The release probability of neuron 1 [subject to alteration by the neighboring astrocyte(s)] is modelled by different gamma distribution for three different conditions: 'control', 'AP5' representing narrowed release probability distribution, and 'AP5-with-control mean' in which the best fit distribution for the AP5 case has been obtained by fixing the mean to remain the same as the mean of the control distribution. (**B**) Summary of simulation illustrating the efficacy with which long-term synaptic plasticity is triggered as a function of the amplitude of postsynaptic current injection $A_2$. Changes in synaptic weight $w$(t) after giving 10 spike pairs at 20 Hz in which the current injection to the presynaptic neuron is 500 pA and the postsynaptic neuron is $A_2$, in control (left), in AP5 (middle), and in AP5-with-control mean (right). The current injection sequences to the presynaptic and the postsynaptic neurons are separated by a time difference $\Delta t$. Color intensity shows the degree of potentiation (red) and depression (blue) of synaptic weight changes observed for variable amount of $A_2$ injected to the postsynaptic neuron at each $\Delta t$ for the three different conditions represented by each panel. Horizontal green dashed line shows the minimum amount of current injection to the postsynaptic neuron $A_2$ that is required to trigger long-term potentiation (LTP) in the control condition, and similarly, the small arrows in each panel indicate the minimum amount of current injection $A_2$ that evokes LTP in control (left), in AP5 (middle), and in AP5-with-control mean (right). Gray-shaded area is the range of $A_2$ used to calculate the average of the maximum change in $w$(t) among $\Delta t$, which is shown in (**C**). For release probability distributions with the same mean, a smaller amount of current injection is needed to trigger LTP if the release probability distribution has a wider variance. (**C**) The averages of the maximum change in $w$(t) in the three different conditions.

The online version of this article includes the following figure supplement(s) for figure 3:

**Figure supplement 1.** Additional data and corresponding fits to show how the effect on the variability of release probability distribution [subject to alteration by the neighboring astrocyte(s)] can be modelled by gamma distributions.

*Figure 3 continued on next page*

*Figure 3 continued*

**Figure supplement 2.** Additional data and comparisons showing how the model can fit experimentally obtained EPSC peak responses recorded in CA1 pyramidal neurons.

evidence for the expression of NMDAR in hippocampal astrocytes and to clarify the relevant NMDAR subtype, we performed single cell RT-PCR to compare the expression of *Grin1*, *Grin2a*, *Grin2b*, and *Grin2c* mRNAs in CA1 astrocytes across SO, SR and SLM layers and in CA1 pyramidal neurons (**Figure 5A**). Acute hippocampal slices were prepared from adult mice as used for electrophysiology experiments, and after labelling astrocytes with sulforhodamine101 (**Nimmerjahn et al., 2004**), RNA

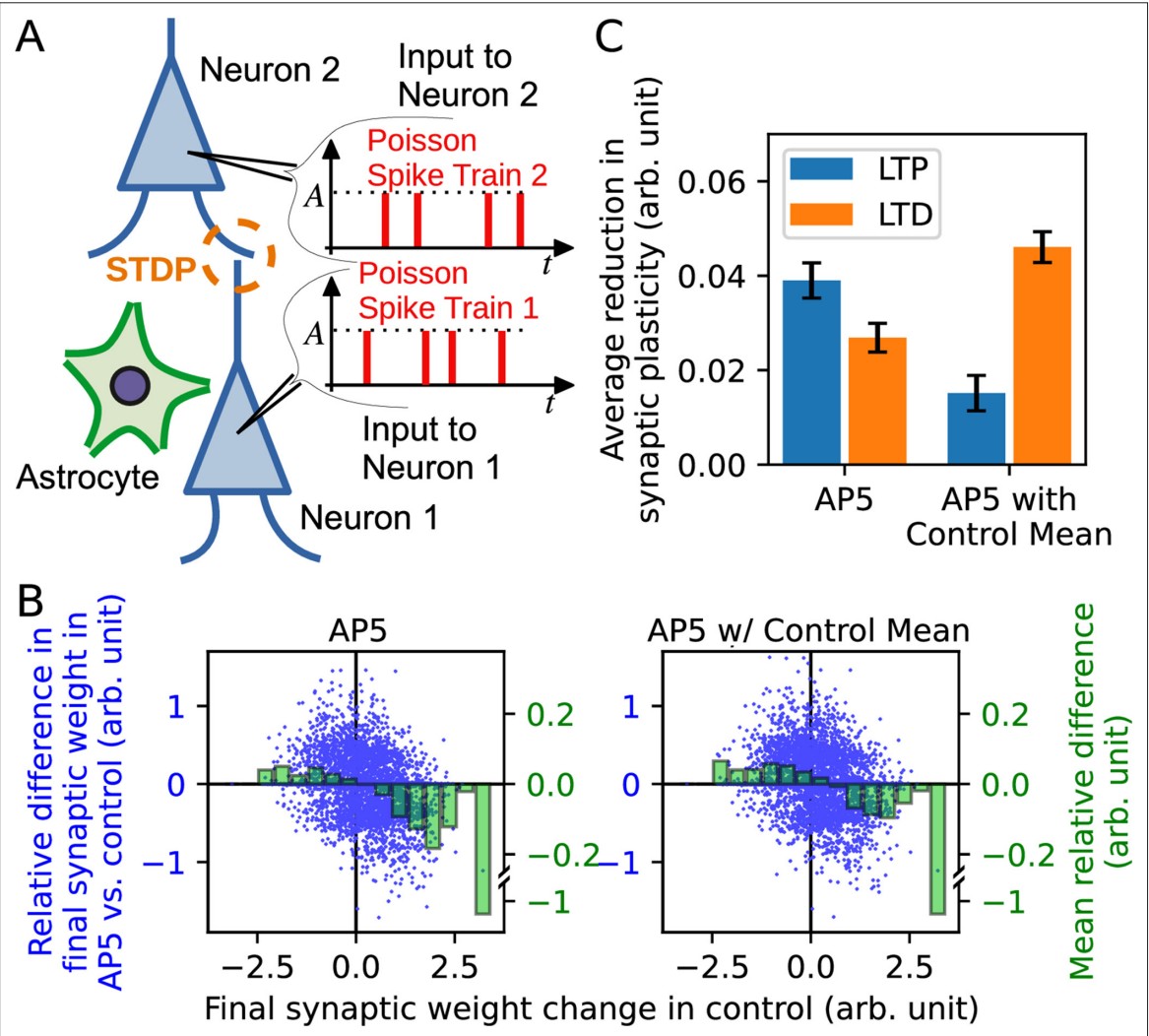

**Figure 4.** Synaptic neurotransmitter release probability diversity impacts long-term synaptic plasticity triggered by Poisson spike input series. (**A**) Illustration of the model used in the simulation. Two leaky integrate-and-fire neurons with a synaptic connection received independent Poisson spike trains. Similarly to **Figure 3**, the release probability of neuron 1 [subject to alteration by the neighboring astrocyte(s)] is modelled by three different gamma distributions representing control, AP5, and AP5-with-control mean conditions. (**B**) Summary of simulation. Major vertical axis (left, blue) shows the relative differences in the final synaptic weights in AP5 (left plot) or AP5-with-control mean (right plot) compared to the control condition. The relative differences (blue dots) are plotted against the final synaptic weight change in control condition (horizontal axis). The binned data are shown as green bars (minor vertical axis: right, green). The plots indicate that spike inputs that promote potentiation under control condition tend to favor weaker potentiation in either AP5 or AP5-with-control mean, and spike inputs that promote depression under control condition tend to favor weaker depression in either AP5 or AP5-with-control mean. (**C**) The reduction of long-term plasticity in AP5 and AP5-with-control mean representing situations of narrowed release probability variance, relative to the control condition. The reduction of plasticity is defined by the mean of the differences between the final weights in NMDAR-blocked conditions and the control condition with a sign correction since long-term potentiation and long-term depression change weights in opposite directions.

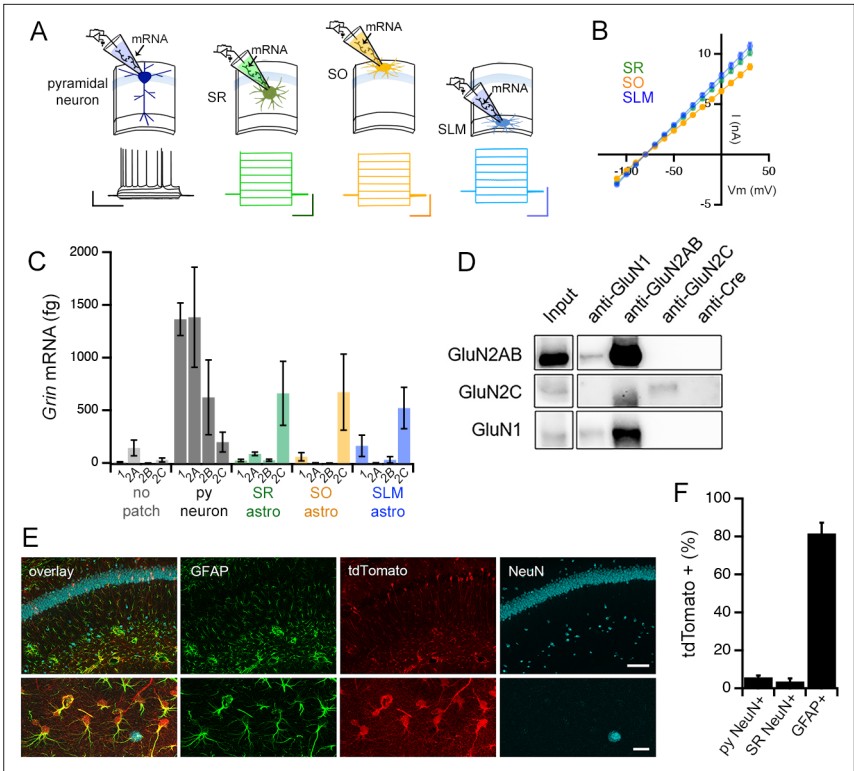

**Figure 5.** Adult mouse hippocampal astrocytes express GluN2C-NMDA receptors. (**A**) Top: patch RT-PCR strategy. Bottom: representative voltage responses to current steps in CA1 pyramidal neuron (far left, black), and current responses to voltage steps in astrocytes in *stratum radiatum* (SR, green), *stratum oriens* (SO, yellow), and *stratum lacunosum moleculare* (SLM, blue). Scale bars: neuron, 40 mV, 500 ms; astrocytes, 6 nA, 200 ms. (**B**) Summary of I-V curves of astrocytes obtained before collecting RNA. N = 39 cells, 36 slices, 9 mice for each layer. (**C**) Summary of RT-PCR analysis for mRNA encoding indicated NMDAR subunits: *Grin1* (1), *Grin2a* ( 2A), *Grin2b* (2B), and *Grin2c* ( 2C). Control samples (no patch) were obtained by inserting the electrode into the slice without patching cells. Three technical replicates were performed for each condition. (**D**) Western blots of co-immunoprecipitations performed using adult mouse hippocampal extracts with the indicated antibodies where anti-Cre antibody is used as a negative control. The blots are probed for GluN2A/2B (top row), GluN2C (middle row) or GluN1 (bottom row). (**E**) Representative brain sections of *Grin2c*^iCre/+ mice crossed to a tdTomato reporter line (Ai9), showing hippocampal area CA1 immunolabelled for GFAP, tdTomato and NeuN; scale bars, 160 µm (top), 25 µm (bottom). (**F**) Quantification of % cells that are positive for tdTomato amongst the counts of NeuN-labelled cells in *stratum pyramidale* (left, total n = 3155 cells) and SR (middle, total n = 619 cells), and GFAP-labelled cells (right, total n = 3149 cells). N = 10 fields of view of area CA1 from individual hippocampal sections (n = 10) obtained from 5 mice. Data are presented as mean ± s.e.m.

from single cells was extracted by patch-clamping. All putative astrocytes had low input resistances (SR 11.20 ± 0.36 MΩ; SO 13.24 ± 0.43 MΩ; SLM 10.60 ± 0.33 MΩ; n = 39 cells from 9 mice sampled for each layer), lacked action potentials, and displayed linear current-voltage relationships (*Figure 5A and B*). CA1 pyramidal neurons expressed high levels of *Grin1*, *Grin2a,* and *Grin2b* subunit mRNAs but expressed substantially low levels of *Grin2c* mRNA. In contrast, astrocytes in all three layers showed robust expression of *Grin2c* mRNA. Moreover, astrocytes showed minimal expression of *Grin2a* and *Grin2b* mRNAs, and the *Grin1* mRNA expression in astrocytes was also substantially low compared to neurons (*Figure 5C*). *Grin2d* mRNA in CA1 pyramidal neurons and astrocytes was also tested, and it was undetectable in RNA extracted from patch clamping both cell types while the probe itself could detect *Grin2d* mRNA in brain tissue extracts (data not shown). Inability to detect *Grin2d* mRNA in astrocytes is consistent with previous reports of single cell RNA-seq analysis (Khakh Lab database; also see *Alsaad et al., 2019*). We also examined the levels of expression of NMDAR subunits in CA3 pyramidal neurons, and the pattern of subunit expression mirrored the pattern observed for CA1 pyramidal neurons (data not shown).

The expression of GluN2C subunit protein in the hippocampus was confirmed by immunoprecipitation of mouse hippocampal extracts followed by Western blotting, using an antibody against the Cre recombinase that is not expressed in wild type mice, as a negative control (*Figure 5D*). For characterizing the cellular expression pattern of GluN2C in the hippocampus, we were unable to identify an antibody against GluN2C that was suitable for immunofluorescence labelling experiments. Therefore, in order to visualize the cells that expressed GluN2C, we used a GluN2C mutant mouse line carrying an insertion of a codon-improved Cre recombinase immediately downstream of the translation initiation site in the *Grin2c* gene (*Miyazaki et al., 2012*). *Grin2c^{iCre/+}* mice were crossed with a Cre reporter line (Ai9: *Madisen et al., 2010*) that expressed tdTomato upon Cre-mediated recombination. The hippocampus of the offspring showed robust tdTomato fluorescence in astrocytes identified by GFAP labelling (81.6% ± 5.7% of GFAP+ ve cells; *Figure 5E and F*), which was consistent with the high expression level of GluN2C mRNA in astrocytes (*Figure 5C*). Moreover, some interneurons in SR identified by the NeuN labeling also expressed tdTomato (3.6% ± 1.6% of NeuN+ ve cells in SR, *Figure 5E and F*), which was in agreement with previous reports (*Ravikrishnan et al., 2018*; *Gupta et al., 2016*). Surprisingly, the pyramidal cell layer also showed sporadic tdTomato-labeled neuronal cell bodies, although such cells represented a limited proportion of pyramidal neurons (5.8% ± 1.0% of NeuN+ ve cells in *stratum pyramidale*, *Figure 5E and F*). The RT-PCR, biochemical, and immunohistochemistry experiments collectively provide evidence in support of GluN2C-containing NMDAR as a major NMDAR type expressed in hippocampal CA1 astrocytes.

## Inhibition of GluN2C NMDARs reduces PPR variability in a manner sensitive to astrocyte GluN1 expression

Given the prominent expression of GluN2C in astrocytes, we wondered if the regulation of PPR could be attributed to GluN2C NMDARs. To test such a possibility, we examined whether pharmacological inhibition of GluN2C NMDARs could mimic the effects of bath applied AP5 and MK801 on PPR variance using QNZ46 (25 μM), an NMDAR antagonist specific for GluN2C/D (*Hansen and Traynelis, 2011*). Similarly to MK801 and APV, QNZ46 reduced the population variance of PPRs without altering the mean PPR (*Figure 6A*; $\delta^2 P = 0.008$; $\bar{x}P = 0.312$). Again we observed the negative correlation between the change in PPR and the baseline PPR (*Figure 6A*, bottom), with a linear fit to the data

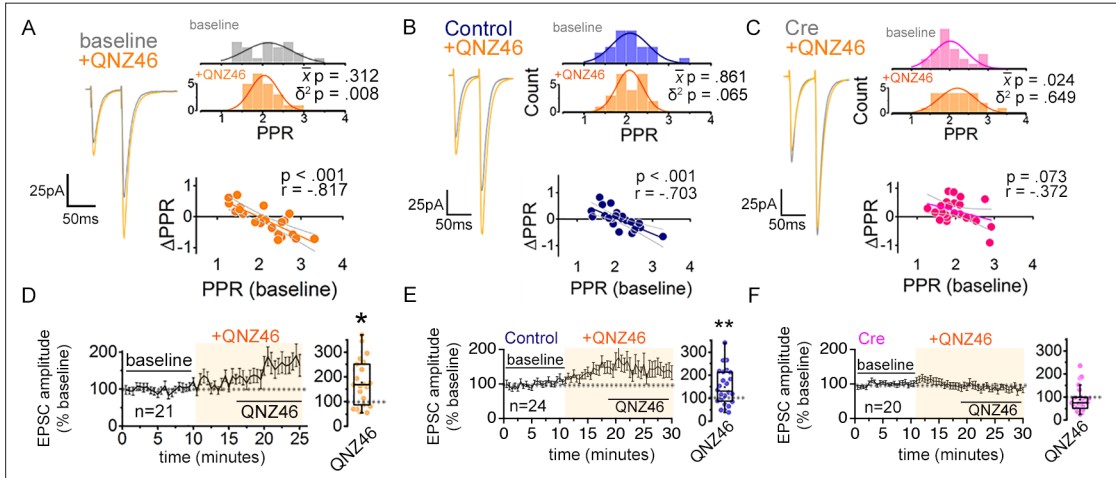

**Figure 6.** Inhibiting GluN2C NMDARs reduces PPR variability in a manner dependent on astrocyte expression of GluN1. (A–C) Left: representative EPSC traces (average of 20 sweeps) to pairs of pulses (Δt = 50 ms) applied to Schaffer collateral axons in baseline and after wash-on of QNZ46 (25 μM). Right: PPR histograms (top) and ΔPPR vs. baseline PPR plots (bottom) before and after bath applying QNZ46 to control uninfected (A), Control (B) and Cre (C) slices. The p values were obtained for the population mean ($\bar{x}$) (paired sample t-test) and variance ($\delta^2$) (one-tailed f-test for equal variances) comparing PPRs before and after the drug treatment. Linear fits and Pearson's correlation coefficients (r) and p values are shown in scatter plots. (D–F) Plots of EPSC amplitudes (normalized to the baseline average) before and during the application of QNZ46 to control uninfected (D), Control (E) and Cre-infected (F) slices (shaded area) where n is the number of inputs examined for each experiment. Baseline and experimental periods are indicated (black bars). Box plots show the summary of normalized EPSC amplitudes during the QNZ46 application. *p < 0.05, **p = 0.005, Mann-Whitney U-test was used to compare the control to the drug conditions. No virus infection+ QNZ46 = 21 inputs, 11 cells, 7 mice; Control+ QNZ46 = 24 inputs, 12 cells, 5 mice; Cre+ QNZ46 = 20 inputs, 12 cells, 6 mice. Data are presented as mean ± s.e.m.

intercepting the x-axis near the baseline mean PPR (X intercept = 2.07). Additionally, in experiments monitoring two independent Schaffer collateral inputs, bath application of QNZ46 decreased the PPR disparity as found for AP5 and MK801 (*Figure 1—figure supplement 5F*; p = 0.005). These observations suggest that GluN2C/D-containing NMDARs contribute to maintaining the broad PPR diversity. Given the lack of detectable expression of GluN2D in hippocampal CA1 cells, it is likely that NMDARs containing GluN2C mediate the observed effects of QNZ46 in modulating presynaptic strength to decrease the overall range of PPR variability.

When EPSC amplitudes were examined, QNZ46 bath application caused a significant increase, reminiscent of the changes observed upon AP5 and MK801 application (*Figure 6D*; p = 0.014). Moreover, similarly to AP5, QNZ46 significantly increased $CV^{-2}$ of EPSC amplitudes (*Figure 1—figure supplement 3B*; p = 0.009), which was suggestive of presynaptic alterations.

Next we asked whether QNZ46 acted on astrocyte NMDARs to control PPR variability. To address this point, we again tested for occlusion using astrocyte GluN1 knock-down slices. QNZ46 was bath applied to Cre and Control AAV-infected slices while recording EPSCs from CA1 neurons infused with MK801. In control slices QNZ46 decreased the PPR variance (*Figure 6B*; p = 0.065) and caused a variable increase in EPSC amplitudes (*Figure 6E*; p = 0.005) without a concomitant change in the amplitude or the frequency of spontaneous EPSCs (*Figure 2—figure supplement 2F*). Crucially, as observed for AP5, QNZ46 was no longer effective in producing a significant change in the PPR variance in Cre slices (*Figure 6C*; p = 0.649). When PPR disparity across two inputs was monitored, QNZ46 decreased the PPR disparity in a manner that was sensitive to astrocyte GluN1 knock-down, which also supported the involvement for GluN2C-NMDARs in regulating presynaptic strength diversity (*Figure 2—figure supplement 2J*,K; Control + QNZ46 p = 0.031; Cre + QNZ46 p = 0.126). Moreover, similarly to AP5, QNZ46-dependent increase of EPSC amplitudes was attenuated also in Cre slices (*Figure 6F*).

Together, these observations indicate that GluN2C NMDARs expressed in astrocytes function to maintain the broad variability of presynaptic strengths across a synapse population. While GluN2C NMDARs also play a role in regulating EPSC amplitudes, this is likely to involve a mechanism that is distinct from the presynaptic regulation but is engaged in parallel to target postsynaptic processes.

## Layer-specific regulation of synaptic strength by astrocyte NMDARs

Astrocytes are a highly heterogeneous cell type (*Zhang and Barres, 2010*; *Khakh and Sofroniew, 2015*) that influence synaptic transmission in a synapse- and circuit-specific manner (*Martin-Fernandez et al., 2017*; *Schwarz et al., 2017*; *Martín et al., 2015*; *Lanjakornsiripan et al., 2018*). Our RT-PCR analysis of NMDAR subtype expression showed relatively high expression of GluN2C mRNA across SR, SO, and SLM layers in CA1 astrocytes compared to CA1 pyramidal neurons. Nevertheless, despite the relatively even expression of GluN2C across layers, our experiments monitoring NMDA-glycine puff application responses in CA1 astrocytes revealed layer-specific differences in the GluN1-dependent component of the slow depolarizing responses triggered by the NMDA-glycine puff, with SR astrocytes showing the highest sensitivity to the astrocyte GluN1 knock-down (*Figure 2—figure supplement 1*). This raised the possibility that astrocyte NMDAR-dependent modulation of synaptic inputs to CA1 pyramidal neurons whose dendrites span across SR, SO, and SLM, might also differ across layers.

To test such possibility of layer-specific regulation, we compared astrocyte NMDAR-dependence of synaptic transmission in CA1 neurons across three layers in slices from astrocyte GluN1 knock-down and control mice. In contrast to the specific decrease in the PPR variance observed in Cre slices relative to Control slices for the SR input (*Figure 2D*), neither the PPR variance nor its mean differed between Cre and Control slices for the SO and the SLM inputs (*Figure 7A–D*; SO: $\delta^2$ P = 0.303, $\bar{x}$P = 0.402; SLM: $\delta^2$ P = 0.624, $\bar{x}$P = 0.161). We also examined the layer-specificity of PPR diversity regulation using a pharmacological approach in slices from wild-type mice. The sensitivity of PPRs to bath applied NMDAR antagonists in CA1 pyramidal neurons intracellularly perfused with MK801, was monitored in response to pairwise stimulation of SO and SLM inputs as had been done for the SR input. Neither MK801 nor AP5 had any appreciable effect on PPR variance in SO (*Figure 7E and G*: MK801, p = 0.107; AP5 p = 0.472) and SLM (*Figure 7I and K*: MK801, p = 0.920; AP5, p = 0.741). MK801 modestly increased EPSC amplitudes at some inputs in SO (*Figure 7F*: p = 0.783) but it had no effect in SLM (*Figure 7J*: p = 0.915). Moreover, AP5 did not potentiate EPSC amplitudes as

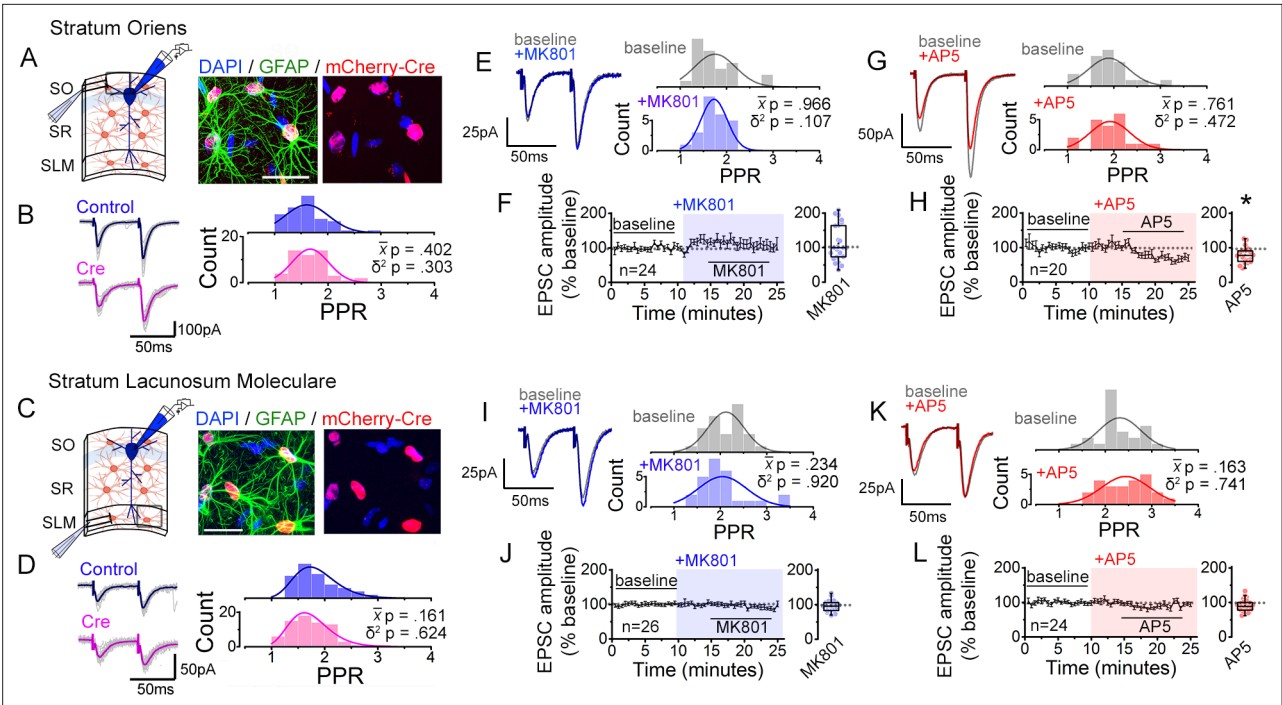

**Figure 7.** Interfering with astrocyte NMDARs has little effect on synaptic strength diversity in the *stratum oriens* (SO) and the *stratum lacunosum moleculare* (SLM). (**A**) Left: recording scheme. Right: confocal image from SO in CA1 showing nls-mCherry-Cre-positive astrocytes labelled with GFAP and DAPI. Scale bar, 20 μm. (**B**) Average (color) and individual (grey) EPSC sweeps from representative recordings by stimulating SO of Control and Cre infected slices from *Grin1*^flx/flx^ mice. PPR histograms from Control (top) and Cre slices (bottom) fit with single Gaussians. [p values: comparisons of population mean ($\bar{x}$) (Mann-Whitney test) and variance ($\delta^2$) (one-tailed f-test for equal variances) between Control and Cre.] Control, n = 50 inputs, 25 cells, 7 mice; Cre, n = 50 inputs, 25 cells, 7 mice. (**C,D**) Same analysis as in *A,B* for SLM. Control, n = 67 inputs, 47 cells, 12 mice; Cre, n = 71 inputs, 49 cells, 12 mice. (**E,G**) Representative EPSC traces and PPR histograms of from SO stimulation in baseline and after applying MK801 (**E**) or AP5 (**G**). (**F,H**) Normalized EPSC amplitudes before and after applying MK801 (**F**) or AP5 (**H**) (shaded area) in SO. Right, summary bar graph. MK801, n = 24 inputs, 12 cells, 6 mice; AP5, n = 20 inputs, 10 cells, 7 mice. * p < 0.05 Mann-Whitney U-test. (**I,K**) Same analysis as in *E,G* for PPRs elicited in SLM in control and MK801 (**I**) or AP5 (**K**). (**J,L**) Same analysis as in *F,H* with MK801 (**J**) or AP5 application (**L**). MK801: n = 26 inputs, 13 cells, 8 mice; AP5, n = 24 inputs, 12 cells, 6 mice. Data are presented as mean ± s.e.m.

The online version of this article includes the following figure supplement(s) for figure 7:

**Figure supplement 1.** MK801 and AP5 do not alter PPR disparity in the *stratum oriens* (SO) or the *stratum lacunosum moleculare* (SLM).

observed for the SR input, but instead it significantly depressed EPSC amplitudes in SO (*Figure 7H*: p = 0.028) while it caused no appreciable change in SLM (*Figure 7L*: p = 0.317). Collectively, these results suggest that regulation of PPR diversity by astrocyte NMDARs in CA1 pyramidal neurons is specific to the SR inputs.

As a further test of the layer specificity of astrocyte NMDAR-dependent regulation of PPR variability, we assessed PPR disparity by performing two independent pathway analysis in each layer in response to bath applied NMDAR antagonists. Again, in contrast to SR input (*Figure 1—figure supplement 5D*, E), neither MK801 nor AP5 depressed the PPR disparity of the SO and the SLM inputs (*Figure 7—figure supplement 1*). These observations are consistent with the role for astrocyte NMDARs, and particularly GluN2C NMDARs, for maintaining the diversity of PPR specifically of SR inputs to CA1 pyramidal neurons.

## Discussion

Broad heterogeneity in the efficacy of synaptic transmission is a fundamental feature of small glutamatergic synapses in the mammalian brain (*Dobrunz and Stevens, 1997*; *Atwood and Karunanithi, 2002*; *Branco and Staras, 2009*). A variety of presynaptic parameters contribute to the observed variability, such as the abundance, subtype, and location of presynaptic calcium channels (*Thalhammer et al., 2017*; *Brenowitz and Regehr, 2007*; *Éltes et al., 2017*), their positioning with respect to

the synaptic vesicles, the active zone size and the number of docked vesicles, the state of vesicle fusion machinery (*Park et al., 2012*; *Marra et al., 2012*; *Fulterer et al., 2018*; *Glebov et al., 2017*; *Holderith et al., 2012*), as well as the coupling to neuromodulatory signals (*Burke et al., 2018*), and the synapse's recent history of synaptic plasticity. These determinants of presynaptic release efficacy are influenced by target-specific signals (*Éltes et al., 2017*; *Branco et al., 2008*; *Reyes et al., 1998*; *Markram et al., 1998*) and are further subject to activity-dependent modulation (*Thalhammer et al., 2017*; *Goda and Stevens, 1998*; *Burke et al., 2018*). Although the basis for presynaptic release probability regulation at individual synapses have been intensively studied, whether and how basal release probability variability is collectively controlled across a synapse population is poorly understood, despite its implications in information processing and memory storage (*Barbour et al., 2007*; *Buzsáki and Mizuseki, 2014*; *Bromer et al., 2018*; *Rotman and Klyachko, 2013*). Our study highlights the novel contribution of astrocyte GluN2C NMDAR signaling in broadening the range of basal release efficacy of Schaffer collateral synapses by effectively maintaining strong synapses stronger and weaker synapses weaker without an overt effect on the mean synaptic strength.

The present study took advantage of paired-pulse response as a measure of presynaptic efficacy. While PPR – a form of short-term plasticity – is inversely related to presynaptic release probability (*Dobrunz and Stevens, 1997*), it does not necessarily always represent changes in presynaptic release across all conditions. For example, PPR changes can arise from a postsynaptic mechanism involving changes in the extent accumulation of desensitized AMPA receptors at the postsynaptic density (*Heine et al., 2008*; *Frischknecht et al., 2009*). Therefore, in some experiments, presumed changes in release probability as monitored by PPR were further corroborated by the correlated changes in $CV^{-2}$ (*Malinow and Tsien, 1990*; *Larkman et al., 1992*). This study extends the concept of NMDAR-mediated glial control of PPR (*Letellier et al., 2016*) to include populations of synapses within defined hippocampal subregions in the healthy adult mouse brain. Sampling a small number of synapses by a relatively weak stimulation intensity (i.e. mean EPSC amplitude of ~60 ± 6 pA to Schaffer collateral stimulation in control condition) consistently revealed a component of PPR distribution that was regulated by astrocyte NMDARs. Notably, such modulation could be masked when sampling a large synapse population that yields stable ensemble responses (*Lines et al., 2017*). Altogether, results from (i) pharmacological experiments using subtype-specific NMDAR antagonists, which allowed monitoring the effects of NMDAR inhibition in the same synapse population, (ii) the genetic interference specifically of astrocyte NMDARs, and (iii) the cellular expression analysis of NMDAR subunits, collectively point to GluN2C NMDAR as the major astrocyte receptor that regulates synaptic strength variability of Schaffer collateral inputs. The use of methods that allow for a more direct estimate of release probability in hippocampal CA1 neurons as have been reported recently (e.g. *Grillo et al., 2018*; *Jensen et al., 2021*) will be informative in further pursuing the underlying mechanisms.

## Functional significance of the broad distribution of presynaptic strengths

What might be an advantage in maintaining a highly variable presynaptic efficacy across a synapse population of a given input type? Our mathematical modeling and simulation data indicate that the width of release probability distribution can bias the outcome of activity-dependent synaptic plasticity, and in a generalized spiking neuron model, a larger release probability variance promoted LTP and LTD. Notably, several studies have implicated astrocyte signaling in LTD in hippocampal CA3 to CA1 synapses (*Chen et al., 2013*; *Andrade-Talavera et al., 2016*; *Navarrete et al., 2019*; *Pinto-Duarte et al., 2019*), where spike timing-dependent LTD in slices from young mice is sensitive to inhibitors of GluN2C/D, although the source of GluN2C/D remains to be determined (*Andrade-Talavera et al., 2016*). Astrocyte GluN2C NMDAR-dependent maintenance of variable basal presynaptic strengths of CA3 to CA1 synapses that we have identified here could therefore be linked to astrocyte signaling that promotes LTP and LTD, which in turn, are not only important for learning and memory but also for network stability (*Collingridge et al., 2010*; *Royer and Paré, 2003*; *Zenke and Gerstner, 2017*). Curiously, this GluN2C NMDAR-dependent broadening of the basal presynaptic efficacy in CA1 neurons is confined to the SR inputs and not observed for SO or SLM inputs. This suggests that for dendritic computations in CA1 pyramidal neurons, the broad variability of synaptic strengths at CA3-CA1 synapses is more crucial compared to the synaptic strength variability of basal or apical tuft inputs. Mice deficient in GluN2C, while mostly normal in their behavior, show deficits

in acquisition of conditioned fear and working memory and changes in neuronal oscillations (*Hillman et al., 2011*; *Mao et al., 2020*). Some population of interneurons express GluN2C, however (*Gupta et al., 2016*; *Ravikrishnan et al., 2018*), and this confounds the interpretation of the observed effects solely to deficits in astrocyte GluN2C. It would be of interest to determine in the future whether there is a learning performance deficit in mice specifically deficient in astrocyte GluN2C or GluN1 in the hippocampal CA1 subfield.

The mechanism by which astrocyte GluN2C NMDARs are coupled to the changes in presynaptic efficacy remains to be clarified. The narrowing of the range of PPR upon astrocyte NMDAR inhibition suggests a capacity for astrocytes to impose dual modulatory actions downstream of astrocyte GluN2C NMDARs, in one possibility, involving the release of two types of gliotransmitters with one potentiating and another depressing synaptic transmission (*Schwarz et al., 2017*; *Covelo and Araque, 2018*). In some cases, the release of a single molecule may also suffice: astrocyte-mediated release of ATP, which is converted to adenosine by extracellular ATPases, has been implicated in the bi-directional modulation of presynaptic efficacy (*Panatier et al., 2011*; *Pascual et al., 2005*; *Zhang et al., 2003*; *Tan et al., 2017*). The effects of adenosine depend in part on the presynaptic $A_{2A}$ or $A_1$ receptors that either enhance or suppress presynaptic function, respectively (*Panatier et al., 2011*; *Tan et al., 2017*). Importantly, a potential mechanism that expands the range of presynaptic efficacy would favor potentiation of strong synapses and depression of weak synapses. At the neuromuscular junction, where glial cells mediate synaptic competition involving glial purinergic signaling, it has been shown that a stronger synaptic input is preferentially potentiated further by the activation of presynaptic $A_{2A}$ receptors while the weaker one remains unchanged or slightly depressed (*Darabid et al., 2018*). Such a mechanism could provide a framework for deciphering the actions of GluN2C NMDARs, where for example, the relative abundance of $A_{2A}$ or $A_1$ receptors at individual presynaptic boutons could determine the polarity of presynaptic efficacy change upon accumulation of extracellular adenosine by the astrocyte GluN2C NMDAR activity, provided that the relative abundance and/or the type of adenosine receptors at individual boutons show correlation to basal release probability. In another scenario, astrocyte GluN2C receptors may influence the release of other gliotransmitters such as glutamate (*Jourdain et al., 2007*) to target presynaptic glutamate receptors and/or GluN2C receptor signaling may influence astrocyte-mediated $K^+$ clearance to locally shape presynaptic action potential waveforms (*Cui et al., 2018*) to in turn affect presynaptic efficacy. Further understanding of such mechanisms is warranted to identify the basis for how the relative differences in synaptic strengths are promoted to facilitate broadening of the range of presynaptic efficacy.

## Astrocyte NMDAR subunit mRNAs and implications for layer-specific synapse modulation

Previous transcriptome analysis of the major cell types in mouse cerebral cortex have suggested that the GluN2C NMDAR subunit mRNA is one of the highly enriched transcripts in astrocytes, whose level can be up to 70-fold of the level found in neurons (*Zhang et al., 2014*). Our single-cell RT-PCR analyses also show robust expression of GluN2C mRNA in astrocytes across the three CA1 layers in contrast to GluN2C mRNA expression in pyramidal neurons. Curiously, despite the high expression of GluN2C mRNA in astrocytes, only low levels of GluN1 mRNA is detected in astrocytes. This finding is unexpected given that GluN1 subunit is required for the surface expression of functional NMDARs (*Fukaya et al., 2003*; *Abe et al., 2004*). Moreover, the observed occlusion of QNZ46 effects on synaptic transmission by astrocyte-specific knock-down of GluN1 also support the presence of functional heteromeric GluN1/GluN2C NMDARs in astrocytes. The whole-cell patch clamp method we used to collect RNAs is biased towards sampling of transcripts that are abundant in the cell body. Given that astrocyte processes are numerous and thin, the discordance between the detected GluN1 and GluN2C mRNA levels could be explained if the mRNAs are differentially localized, with GluN1 mRNA being preferentially targeted to processes compared to the cell body of astrocytes. Such a proposal is consistent with a recent study reporting of GluN1 mRNA in astrocyte processes that is locally translated (*Sakers et al., 2017*). The precise intracellular localization of NMDAR subunit mRNAs in astrocytes remains to be determined.

The present study revealed synapse regulation by the astrocyte GluN2C NMDARs that is confined to the SR input although GluN2C mRNA is expressed broadly across CA1 astrocyte layers. Therefore, the layer-specificity of synaptic modulation could potentially arise from features of GluN2C NMDAR

assembly and trafficking and/or signaling that is unique to SR astrocytes over SO and SLM astrocytes. In support of layer-specific differences in astrocyte NMDAR signaling, the slow astrocyte membrane depolarization triggered by puff applied NMDA-glycine shows significant dependence on astrocyte NMDAR only in SR and not in SO nor in SLM (*Figure 2—figure supplement 1*). Additionally, differences in the properties of presynaptic inputs to CA1 pyramidal neuron dendrites across layers (e.g. *Schroeder et al., 2018*) could also contribute to layer-specific synaptic modulation by astrocyte GluN2C NMDARs.

The GluN2C subunit forms diheteromeric receptor complexes with the obligate GluN1 subunit and triheteromeric receptor complexes with GluN1 and GluN2A (*Paoletti et al., 2013*; *Hansen et al., 2018*). The presence of GluN2C confers NMDAR properties that are distinct from NMDARs containing GluN2A or GluN2B which are abundant in neurons. For example, GluN2C-containing NMDARs show reduced channel open probability, increased glutamate sensitivity, slow receptor deactivation, and a decreased sensitivity to $Mg^{2+}$ block (*Paoletti et al., 2013*; *Hansen et al., 2018*). Weak $Mg^{2+}$ binding would enable open-pore blockers such as MK801, to nonetheless rapidly exert their inhibitory action. Such reduced $Mg^{2+}$ sensitivity of astrocyte GluN2C NMDARs could have contributed to the rapid effect observed for MK801 in normalizing presynaptic strengths and reducing the PPR disparity. Moreover, the low $EC_{50}$ of GluN2C-containing NMDAR activation to glutamate (*Hansen et al., 2018*) suggests that they may be well suited for detecting synaptic release events at perisynaptic astrocyte processes that can be at some distance from the active zone. Several drugs that target NMDARs have been in clinical use, such as ketamine as an anesthetic and treatment for depression (*Krystal et al., 2019*; *Williams and Schatzberg, 2016*) and memantine for the treatment of moderate to severe dementia in Alzheimer's disease (*Graham et al., 2017*). Under physiological conditions, GluN2C/GluN2D-containing NMDARs display up to 10-fold higher sensitivity to ketamine and memantine in comparison to GluN2A/GluN2B-containing NMDARs that are highly expressed in neurons (*Kotermanski and Johnson, 2009*; *Hansen et al., 2017*). Therefore, although therapeutics targeting NMDARs to date have largely focused on neuronal NMDARs, it would be crucial to consider also the consequences of interfering with GluN2C NMDARs that are enriched in astrocytes, which is underscored by the increasing recognition of the involvement of astrocytes in a variety of neurological disorders (*Zuchero and Barres, 2015*; *Chung et al., 2015*).

# Materials and methods

## Slice preparation and whole cell recordings

Mice (P60-120) were deeply anesthetized with isoflurane and transcardially perfused with ice-cold cutting aCSF containing (in mM) 93 N-methyl D-glucamine, 2.5 KCl, 1.2 $NaH_2PO_4$, 30 $NaHCO_3$, 20 HEPES, 20 glucose, 5 Na ascorbate, 2 thiourea, 3 sodium pyruvate, 12 N-acetyl L-cysteine, 10 $MgSO_4$, 0.5 $CaCl_2$, pH adjusted to 7.4 with HCl and bubbled with 95 % $O_2$/5 % $CO_2$. Brains were extracted and 350 μm transverse hippocampal slices were cut in ice-cold cutting aCSF on a Leica VT1200 vibrating microtome. Slices were incubated for 12 min in cutting aCSF warmed to 34 °C, then placed in holding solution containing (in mM) 81.2 NaCl, 2.5 KCl, 1.2 $NaH_2PO_4$, 30 $NaHCO_3$, 20 HEPES, 20 D-glucose, 5 Na ascorbate, 2 thiourea, 3 sodium pyruvate, 12 N-acetyl L-cysteine, 2 $MgSO_4$, 2 $CaCl_2$, pH 7.4, bubbled with 95 % $O_2$/5 % $CO_2$ for up to 8 h. In some cases, slices were incubated in 50 μM sulphorhodamine for 30 min at 34 °C to identify astrocytes for RNA extraction.

Whole-cell patch clamp recordings were obtained from CA1 pyramidal neurons or astrocytes using an Olympus BX51W1 microscope equipped with IR-DIC optics and a motorized stage. Images were captured with a digital camera (Andor, iXon3) and imaged using MetaMorph software (Molecular Devices). Voltage clamp and current clamp experiments were carried out using Multiclamp 700B amplifiers (Molecular Devices) and up to four micromanipulators (SM5, Luigs and Neumann). Patch pipettes (tip resistance 2–4 MΩ for neurons, 4–6 MΩ for astrocytes) were pulled using a vertical 2-stage puller (PC-10 Narishige). Slices were constantly perfused with recording aCSF containing (in mM) 119 NaCl, 2.5 KCl, 1.3 $NaH_2PO_4$, 26 $NaHCO_3$, 1 $MgCl_2$, 2 $CaCl_2$, 20 D-glucose and 0.5 Na ascorbate pH 7.4, bubbled with 95 % $O_2$/5 % $CO_2$ and maintained at 32–34 °C using an in-line heater (Harvard Instruments). Picrotoxin (100 μM) was added to the recording aCSF to block $GABA_A$-receptors and isolate glutamatergic synaptic transmission. Internal pipette solution contained (in mM) 130 $CsMeSO_3$, 8 NaCl, 4 Mg-ATP, 0.3 Na-GTP, 0.5 EGTA, 10 HEPES, pH 7.3, 290–295 mOsm for neuron recordings,

and 130 K gluconate, 10 HEPES, 4 $MgCl_2$, 4 $Na_2$-ATP, 0.4 $Na_3$-GTP, 10 Na-phosphocreatine, pH 7.3, 290 mOsm for astrocyte recordings. 50 µM AlexaFluor488 or AlexaFluor594 hydrazide (ThermoFisher Scientific) was included in the pipette solution to verify cell identity and to position stimulating and puff pipettes. Neurons were voltage clamped at –70 mV and series resistance ($R_s$) was monitored throughout all recordings and left uncompensated. Experiments were discarded if $R_s$ <30 MΩ and/or changed by >20%. Average $R_s$ was 25.24 ± 0.27 MΩ (n = 298 cells).

## Recordings of synaptic transmission

For all experiments using NMDAR antagonists, 1 mM MK801 was included in the patch pipette solution. Internal solution was allowed to equilibrate into the cell for 10–15 min and inputs were stimulated at a low frequency (0.1 Hz) with pairs of pulses at least 45 times before beginning the experiment to pre-block postsynaptic NMDAR receptors. Effective inhibition of NMDAR currents by internal MK801 was confirmed in a separate set of experiments (*Figure 1—figure supplement 1*). Pipettes were tip filled with ~0.2 µl of internal solution lacking MK801 to avoid its leakage to the extracellular milieu prior to seal formation. Small bundles of axons in the *stratum radiatum*, *oriens*, or *lacunosum moleculare* were stimulated using AgCl bipolar electrodes in theta-glass pipettes (tip diameter ~2–3 µm) filled with recording aCSF, connected to a stimulus isolation unit (A360, WPI). These bundles are identified throughout as 'input'. Stimulation strengths varied between ~50 and 500 µA and were adjusted to obtain EPSCs of approximately 50–100 pA. Up to three independent inputs were sampled during a single experiment, although a maximum of two independent inputs were activated per input pathway. When multiple stimulation electrodes were used they were positioned on opposite sides of the neuron and/or in different input pathways (i.e. SR and SLM or SR and SO). When sampling two inputs in a single pathway, the independence of the two inputs was confirmed by a cross-paired-pulse stimulation paradigm (*Otani and Connor, 1996*; *Scimemi et al., 2004*); after stimulating one of the two inputs the other input was stimulated in quick succession (50 ms inter-pulse interval). If facilitation or depression was observed in the second pulse, then the position of stimulation electrode was changed and independence of the two inputs was re-assessed. During the experiment, EPSCs from each pathway were sampled with paired pulses every 30 s. Inputs from separate pathways were stimulated at least 5 s apart. EPSCs were sampled over a 10 min baseline period (i.e. 20 sweeps) before the perfusion of drugs and continued for at least an additional 20 min. EPSC amplitudes were averaged over 20 pre-drug baseline sweeps, and 20 post-drug sweeps. The change in EPSCs (ΔEPSCs) was calculated as the ratio of the average post-drug amplitude to the average baseline amplitude. PPRs were calculated based on the average EPSCs of 5 sweep bins (i.e. 2.5 min). Baseline and post-drug averages were calculated as the average of four bins (i.e. over 10 min) before and after drug application, respectively. Changes in PPR (ΔPPR) were calculated as the difference between the drug application value and the baseline value. Values for coefficient of variation (CV) were obtained from 20 baseline, and 20 post-drug sweeps. The change in $CV^{-2}$ ($\Delta CV^{-2}$) was calculated as the ratio of post-drug $CV^{-2}$ to pre-drug $CV^{-2}$. EPSCs and sEPSCs amplitudes in the absence of drug application were stable over the duration of the experiment, and sEPSC amplitudes were stable across all drug conditions tested (*Figure 1—figure supplement 3*, *Figure 2—figure supplement 1*), suggesting that run-down (i.e. non-stationarities) will not substantially influence the outcome of the $CV^{-2}$ analysis. All EPSC amplitude, rise-time, and decay measurements were performed in Clampfit 10.6 software. sEPSCs were identified as events outside a 50 ms window following the second stimulation pulse for each pathway using the template matching algorithm in Clampfit 10.6.

## Astrocyte recordings and NMDA/glycine puff

Astrocytes were identified in acute slices based on mCherry or sulphorhodamine fluorescence, or on their appearance under IR-DIC observation (small, circular cell bodies in the neuropil). Astrocyte identity was always confirmed by their passive electrical properties (linear I-V relationship), low input resistance ( < 20 MΩ), and low resting membrane potential (< –75 mV), as well as post-hoc labeling by Alexa dyes included in the patch pipette. Recording aCSF contained picrotoxin (100 µM), tetrodotoxin (0.5 µM), and CNQX (10 µM) to reduce network excitability associated with the application of iGluR agonists. Patch pipettes ($R_t$ = 4–6 MΩ) were used to locally deliver recording aCSF solution containing 1 mM NMDA and 1 mM glycine. Puff pipettes were placed approximately 50 µm from the

patched cell and puff pressure (3 psi, 100 ms duration) was controlled with a Picospritzer III (Parker Hannifin) connected to $N_2$ gas.

## Whole-cell patch RNA extraction from astrocytes and neurons

Whole-cell patch clamp recordings from astrocytes and neurons were performed using pipettes containing (in mM) 130 K gluconate, 10 HEPES, 4 MgCl$_2$, 4 Na$_2$-ATP, 0.4 Na$_3$-GTP, 10 Na-phospho-creatine, 1 U/µl RNAase inhibitor, 50 µM AlexaFluor488, pH 7.3, 290 mOsm. Pipettes were tip filled (~0.2 µl) with the same internal solution but lacking RNAase inhibitor in order to facilitate obtaining of GΩ seals. After determining the electrical properties of the patched cell to confirm its identity, RNA was extracted using a previously published protocol (*Fuzik et al., 2016*). The cell was held at –5 mV, and repetitively depolarized to +20 mV for 5 ms at 100 Hz for ~5 min while light negative pressure was applied to the pipette. The AlexaFluor fluorescence signal was visualized to confirm the extraction of cell cytoplasm.

## Single-cell quantitative PCR

Individual patched cells were processed following the provider´s recommendation for the Single Cell-to-CT qRT-PCR Kit (ThermoFisher Scientific). cDNAs for *Grin1, Grin2a, Grin2b, Grin2c, Grin2d,* and *Rn28s1* were quantified by TAQMAN system using the following probes. The provider's recommended pre-amplification step was performed for all the genes except for *Rn28s1*.

> *Grin1*: fw 5'GACCGCTTCAGTCCCTTTGG 3', rv 5' CACCTTCCCCAATGCCAGAG 3', probe 5' AGCAGGACGCCCCAGGAAAACCAC 3' (MGBNFQ, 6FAM)
> *Grin2a*: fw 5' AGACCCCGCTACACACTCTG 3', rv 5'TTGCCCACCTTTTCCCATTCC 3', probe 5' AGCACGATCACCACAAGCCTGGGG 3' (MGBNFQ, 6FAM)
> *Grin2b*: fw 5'GGCATGATTGGTGAGGTGGTC 3', rv 5'GGCTCTAAGAAGGCAGAAGGTG 3', probe 5'ATTGCTGCGTGATACCATGACACTGATGCC 3' (MGBNFQ, 6FAM)
> *Grin2c*: fw 5'GGAGGCTTTCTACAGGCATCTG 3', rv 5'ATACTTCATGTACAGGACCCCATG 3', probe 5'TCCCACCGTCCCACCATCTCCCAG 3' (MGBNFQ, 6FAM)
> *Grin2d*: fw 5'TCAGCGACCGGAAGTTCCAG 3', rv 5'TCCCTGCCTTGAGCTGAGTG 3', probe 5' TCCTCCACTCTTGGCTGGTTGTATCGCA 3'(MGBNFQ, 6FAM)
> *Rn28s1*: fw 5'CCTACCTACTATCCAGCGAAACC 3', rev 5'AGCTCAACAGGGTCTTCTTTCC 3', probe 5' CTGATTCCGCCAAGCCCGTTCCCT 3' (MGBNFQ, VIC)

RNA levels were normalized by the quantity of *Rn28s1*, and standard curves were prepared for estimating the quantity (in femtograms) of the targeted RNAs after pre-amplification. For making the standard curves, total RNA from mouse brain hippocampal tissue was extracted using TRIZOL reagent. Target cDNAs were amplified by regular PCR, and amplicons were purified, quantity of cDNA was measured and submitted to serial dilutions, used for standard curves in TAQMAN system, in parallel to the single cell derived cDNA samples.

## Intracranial AAV injections

Recombinant AAV vectors were targeted to the dorsal CA1 hippocampus of adult (P60-P90) *Grin1$^{flx/flx}$* mice as previously described (*Letellier et al., 2016*; *Cetin et al., 2007*). Bilateral injections (~600 nl/hemisphere at a rate of 200 nl/min) of AAV9.2-GFAP104-nls-mCherry-Cre (6.67×10$^{13}$ genome copies/ml), AAV9.2-GFAP104-nls-mCherry (i.e. Control; 6.86×10$^{13}$ genome copies/ml), AAVDJ8-GFAP104-nls-mCherry-Cre (5.95×10$^{13}$ genome copies/ml), or AAVDJ8-GFAP104-eGFP (also denoted Control for simplicity; 3.98×10$^{13}$ genome copies/ml) were made using the following coordinates: X (posterior from Bregma) – 1.4 mm, Y (lateral from sagittal suture) ± 1.4 mm, Z (ventral from pia) – 2.1 mm. Viruses were expressed for at least 21 days (max post-surgical duration of 40 days) before mice were used in physiology or imaging experiments. Comparisons were always made between Cre and Control (Cre-lacking) virally infected slices from animals that received the same AAV serotype. While most experiments used the AAV9.2 viruses, the following data used the AAVDJ8: *Figures 2 and 7*, *Figure 2—figure supplement 2A, B*; see E-phys statistics report for details.

## Immunohistochemistry

Slices prepared from *Grin1$^{flx/flx}$* mice expressing GFAP104-nls-mCherry-Cre were fixed in 4 % paraformaldehyde (PFA) in 0.1 M phosphate buffer solution (PB) for 1–2 hr, then washed in PB and stored

overnight. Alternatively, mice brain tissue was fixed with 4 % PFA in PB (pH7.5) by transcardial perfusion, followed by further overnight incubation of the removed brain. Fixed brains were cryosectioned to a thickness of 40 μm. Slices were permeabilized with 0.3 % Triton X-100 and blocked in 10 % goat serum, then incubated in primary antibody overnight at 4 °C. Slices were washed and incubated in secondary antibodies for 1–2 hr at room temperature. Slices were then washed and mounted in ProLong antifade mounting medium (ThermoFisher Scientific) containing DAPI (1:1000, ThermoFisher Scientific), and visualized on a Zeiss 780, or an Olympus FV1200 or FV3000 confocal microscope. The primary antibodies used were chicken anti-RFP (1: 1000, Rockland #600-901-379), mouse anti-GFAP (1:1000, Synaptic Systems #173011), rabbit anti-GFAP (1:500, Abcam ab48050), rabbit anti-NeuN (1:500, Abcam #ab177487), and guinea pig anti-GluN1 (1:100, Alamone Labs AGP-046). The secondary antibodies used were AlexaFluor555 goat anti-chicken (1:1000, Invitrogen #A32932), AlexaFluor633 goat anti-mouse (1:1000, Invitrogen #A21052), AlexaFluor488 goat anti-rabbit (1:1000, Invitrogen #A11034), and AlexaFluor647 goat anti-guinea pig (1:200, Invitrogen A21450).

## NMDA receptor immunoprecipitation from mouse brain

Mouse hippocampal tissues were dissociated using a glass dounce homogenizer in a 50 mM Tris-HCl, pH 9.0 buffer containing protease inhibitors (10% w/v, cOmplete, Merck). Subsequently, 1 % (w/v) sodium deoxycholate was added and incubated for 30 min at 37 °C with mild shaking to solubilize the tissue. Samples were centrifuged at 100,000 rpm at 4 °C for 1 hr. Supernatant was collected, protein concentration was measured by the BCA assay, and stored at –80 °C for later analysis or diluted five-fold in a 50 mM Tris-HCl pH 7.5 buffer containing 0.1 % of Triton X-100 for co-immunoprecipitation experiments.

To test for co-immunoprecipitation, a mix of Sepharose Fast-Flow protein A and protein G beads was prepared. For each antibody reaction, 40 μl of the mixed resin was incubated with 5 μg of antibody. After at least 2 hr of incubation at 4 °C excess antibody was removed by washing, and 2 mg of protein extract was added to each sample containing the resin beads bound by the antibody and incubated overnight at 4 °C. Resins were washed three times with 10 volumes of 50 mM Tris-HCl pH 7.5 with 0.1 % Triton X-100. Supernatant was carefully removed, and resins were suspended in 2 x SDS-PAGE protein loading buffer containing DTT. The unbound protein extract was treated for a second overnight incubation at 4 °C with a freshly prepared antibody-bound resin under identical conditions as the first overnight incubation, to ensure effective pull-down of soluble NMDA receptor content from the extract. Co-immunoprecipitated proteins from the two rounds of incubation were pooled together.

Protein samples (inputs and co-immunoprecipitated proteins) were heat denatured for 2 min at 95 °C and subjected to 6 % SDS-PAGE separation followed by western blot. Antibodies used were: mouse anti-NR1 (Synaptic Systems #114011), rabbit anti-GluN2A/2B (Synaptic Systems #244003), rabbit anti-GluN2C (generously provided by Dr. Masahiko Watanabe or purchased from Frontier Institute #GLURE3C-RB-AF270), and mouse anti-Cre (Merck Millipore clone 2D8).

## FM1-43 experiments for estimating the release probability distributions used for modeling

Release probability distributions for mathematical modelling were obtained by measurements of readily releasable pool using the styryl dye FM1-43 (Invitrogen, Fisher-Scientific) in rat primary hippocampal neurons co-cultured with astrocytes as described previously (*Goda and Stevens, 1998*; *Letellier et al., 2019*). Images were captured on an inverted Olympus IX71 microscope equipped with an EMCCD camera (Andor Technology, Oxford Instruments) controlled by Metamorph software (Molecular Devices). The extracellular solution consisted of (in mM) 137 NaCl, 5 KCl, 10 D-Glucose, 5 HEPES pH 7.3, 2 $CaCl_2$, 2 $MgCl_2$, 0.01 CNQX, 0.1 picrotoxin at 300 mOsm. Briefly, neurons (used at DIV11-14) were stimulated by a pair of field electrodes (positioned ~10 mm apart) using 40 action potentials at 20 Hz in the presence of 10 μM FM1-43 and for an additional minute in FM1-43 but without stimulation to allow for completion of endocytosis. Subsequently, cells were washed in extracellular solution containing 1 mM Advasep-7 (Biotium) for 1 min to facilitate dye removal, and then the wash was continued for a total period of 10 min. Images were acquired before and after the unloading stimulation, which was 600 action potentials at 20 Hz. The signal remaining was taken as background.

The images were analyzed by OpenView software (*Kaufman et al., 2012*) kindly provided by Dr. Noam Ziv.

## Mathematical model for the numerical investigation

We used the synapse model based on the Tsodyks-Pawelzik-Markram model (*Tsodyks et al., 1998*) and the leaky integrate-and-fire neuron model to reproduce ratios of peaks of EPSC waveforms with a 20 Hz spike input. The synaptic dynamics is modeled by [1]:

$$\frac{dx_0(t)}{dt} = \frac{x_2(t)}{\tau_{rec}} - u_0\left(t_{sp}^+\right) x_0\left(t_{sp}^-\right) \delta\left(t - t_{sp}\right) \tag{1}$$

$$\frac{dx_1(t)}{dt} = -\frac{x_1(t)}{\tau_{in}} + u_0\left(t_{sp}^+\right) x_0\left(t_{sp}^-\right) \delta\left(t - t_{sp}\right) \tag{2}$$

$$\frac{dx_2(t)}{dt} = \frac{x_1(t)}{\tau_{in}} - \frac{x_2(t)}{\tau_{rec}} \tag{3}$$

$$\frac{du_0(t)}{dt} = -\frac{u_0(t)}{\tau_f} + U_{SE}\left(1 - u_0\left(t_{sp}^-\right)\right) \delta\left(t - t_{sp}\right) \tag{4}$$

Here, $x_0$ is the portion of available neurotransmitters, $x_1(t)$ is the portion of neurotransmitters released by pre-synaptic spikes, $x_2(t)$ is the portion of neurotransmitters being recovered, $u_0(t)$ is the utilization of available neurotransmitters after each spike. $\tau_{in}$ is the timescale of neurotransmitter release. $\tau_{rec}$ is the recovery timescale. $\tau_f$ is the timescale of synaptic facilitation. $U_{SE}$ is the initial release probability without the influence of synaptic facilitation. In this model, the dynamical variables are $x_i$ with $x_0 + x_1 + x_2 = 1$. $U_{SE}$ is a number determined by a random number drawn from a gamma distribution. Readily releasable pool size, which is monitored by the FM1-43 labeling as a proxy of release probability at individual synapses, is fitted with a gamma distribution (c.f. *Murthy et al., 1997*):

$$P(z; k, \theta) = \frac{1}{\Gamma(k)\theta^k} z^{k-1} e^{-\frac{z}{\theta}} \tag{5}$$

where $z$ is the FM1-43 signal, $k$ is the shape parameter and $\theta$ is the scale parameter. The release probability $U_{SE}$ is then given by $U_{SE} = \frac{z}{140.0}$, by normalizing $z$ that is proportional to release probability by an arbitrarily chosen maximal value of FM1-43 signal, which is at most 140 in our measurements (*Figure 3—figure supplement 1*). The FM1-43 signal distributions in control and AP5 conditions are modeled by different shapes of the distribution. The fits and observed measurements of the FM1-43 distributions are presented in *Figure 3—figure supplement 1*. The best fit parameters for different conditions are $(k, \theta) = (1.40, 26.5)$ for control, $(k, \theta) = (3.12, 10.1)$ for AP5, and $(k, \theta) = (2.94, 12.6)$ for AP5 when the mean is fixed to stay the same as for the control ('AP5 with control mean').

In the simulations, the model is based on leaky integrate-and-fire (LIF) neurons. There is a synaptic connection between neurons. The membrane potential $V_m$ of a LIF neuron is given by

$$C_m \frac{dV_m}{dt} = -\frac{(V_m - E_L)}{R_m} + I_{stim} + I_E \tag{6}$$

where $C_m$ is the membrane capacitance, $R$ is the membrane resistance, $I_{stim}$ is the input current and $I_E$ is the excitatory current triggered by an excitatory input. In this study, we set $C_m = 150$ pF and $R_m = 100$ MΩ, with the time constant $\tau = 15$ ms for the neurons. Additionally, the following values are used: the threshold potential $V_{thre} = -50.0$ mV, reversal potential $E_L = -65.0$ mV, and refractory time $\tau_{ref} = 2$ ms.

The excitatory current $I_E$ is given by

$$I_E = g_E\left(E_E - V_m\right) \tag{7}$$

where $g_E$ is the conductance and $E_E = 0$ mV is the reversal potential of the excitatory response. The dynamics of conductance is given by

$$\tau_{syn,E} \frac{dg_E}{dt} = -g_E + w\left(t_{sp}\right) u_0\left(t_{sp}^+\right) x_0\left(t_{sp}^-\right) \delta\left(t - t_{sp}\right) \tag{8}$$

where $\tau_{syn,E}$ is the time constant of the excitatory synaptic response and $w(t)$ is the weight of the connection. Here we set $\tau_{syn,E}$ to be 3 ms, which is approximately the weighted average decay time constant of AMPA receptors on dendrite and soma of hippocampal CA1 pyramidal neurons (*Spruston et al., 1995*). With this setting, parameters $\tau_f$, $\tau_{rec}$, and $w(0)$ can be determined by fitting both EPSC peaks and PPR variance observed in the experiment (*Figure 3—figure supplement 2*).

The spike-time-dependent plasticity (STDP) used to modify $w(t)$ follows the rule used by *Rubin et al., 2001*:

$$\Delta w\left(t\right) = \begin{cases} \lambda[w_{\max} - w(t)] \exp\left[\frac{(t^{\mathrm{pre}} - t^{\mathrm{post}})}{\tau^{+}}\right] & , \text{ if } t^{\mathrm{post}} > t^{\mathrm{pre}} \\ \lambda w(t) \exp\left[\frac{(t^{\mathrm{post}} - t^{\mathrm{pre}})}{\tau^{-}}\right] & , \text{ if } t^{\mathrm{post}} > t^{\mathrm{pre}} \end{cases}$$

where $w_{max} = 2w\left(0\right)$, $\lambda = 0.1$, and $\tau^{+} = \tau^{-} = 20$ms. $t^{pre}$ and $t^{post}$ are spike times of presynaptic and postsynaptic neuron respectively.

## Statistics

All statistical analyses were performed using OriginPro software (OriginLab Corp.). Datasets were tested for normality using the Shapiro Wilk test. When the criteria for normality was achieved, differences of mean values were examined using paired or unpaired Student's two-tailed t-tests or one-way ANOVAs. When the criteria for normality was not achieved, differences in mean values were examined using Mann-Whitney or Kruskal Wallis tests. Normalized mean values obtained from recordings (i.e. normalized EPSCs) were compared to values obtained at the same time point in control experiments using Mann-Whitney tests. Variance of PPR distributions were examined using one-tailed f-tests for equal variances or Levene's test. Box plots represent median and quartile values, whiskers represent maximum and minimum values that are not outliers. p Values are indicated throughout or indicated by * if p < 0.05. Sample sizes for electrophysiology experiments were determined based on PPR disparity obtained in pilot studies and experimental data outlined in *Letellier et al., 2016*. A power analysis was performed using a $\beta$ value of 0.8 and an $\alpha$ value of 0.05. Standard deviations for PPR disparity data were determined to be ~0.35, and the expected effect size is ~45 %.

## Acknowledgements

We thank all members of the Goda laboratory for providing valuable and ongoing feedback throughout the course of this study. We thank Tatjana Tchumatchenko, Thomas Chater and Toru Shinoe for comments on an earlier version of the manuscript, Noam Ziv for kindly sharing the custom written OpenView software, Yun Kyung Park for help with RNA collection for RT-PCR experiments, Toru Shinoe for pilot electrophysiology experiments, Tom McHugh and Shigeyoshi Itohara for kindly providing *Grin1*flx/flx mice and Ai9 mice, respectively, and Masahiko Watanabe for generously providing GluN2C antibodies. PC was an Overseas Research Fellow of the Japan Society for the Promotion of Science (P14760). This work was supported by the RIKEN Center for Brain Science, the Uehara Memorial Foundation, JSPS Core-to-Core Program (JPJSCCA20170008), Grants-in-Aid for Scientific Research (15H04280, YG; 18H05213, TF; 19K16885, CCAF) from the MEXT, and the Brain/MINDS from the Japan AMED.

## Additional information

### Competing interests

Yukiko Goda: Reviewing editor, eLife. The other authors declare that no competing interests exist.

### Funding

| Funder | Grant reference number | Author |
| --- | --- | --- |
| Japan Society for the Promotion of Science | Overseas Research Fellow (P14760) | Peter H Chipman |
| Japan Society for the Promotion of Science | Core-to-Core Program (JPJSCCA20170008) | Yukiko Goda |

| Funder | Grant reference number | Author |
| --- | --- | --- |
| MEXT Grants in Aid for Scientific Research | 15H04280 | Yukiko Goda |
| MEXT Grants in Aid for Scientific Research | 18H05213 | Tomoki Fukai |
| RIKEN Center for Brain Science | | Yukiko Goda |
| Uehara Memorial Foundation | | Yukiko Goda |
| Japan AMED Brain/MINDS | | Yukiko Goda |
| MEXT Grants in Aid for Scientific Research | 19K16885 | Chi Chung Alan Fung |

The funders had no role in study design, data collection and interpretation, or the decision to submit the work for publication.

## Author contributions

Peter H Chipman, Conceptualization, Data curation, Formal analysis, Funding acquisition, Investigation, Methodology, Validation, Visualization, Writing – original draft, Writing – review and editing; Chi Chung Alan Fung, Data curation, Formal analysis, Investigation, Methodology, Resources, Software, Visualization, Writing – original draft, Writing – review and editing; Alejandra Pazo Fernandez, Abhilash Sawant, Data curation, Formal analysis, Investigation, Methodology, Visualization; Angelo Tedoldi, Data curation, Investigation, Methodology, Visualization; Atsushi Kawai, Investigation, Methodology; Sunita Ghimire Gautam, Investigation, Methodology, Resources, Validation; Mizuki Kurosawa, Investigation, Methodology, Resources; Manabu Abe, Kenji Sakimura, Resources; Tomoki Fukai, Formal analysis, Funding acquisition, Investigation, Methodology, Supervision, Writing – review and editing; Yukiko Goda, Conceptualization, Funding acquisition, Investigation, Project administration, Supervision, Writing – original draft, Writing – review and editing

## Author ORCIDs

Yukiko Goda http://orcid.org/0000-0003-0352-9498

## Ethics

All animal experiments were approved by the RIKEN Animal Experiments Committee and performed in accordance with the RIKEN rules and guidelines. [Animal Experiment Plan Approval no. W2021-2-015(2)].

## Decision letter and Author response

Decision letter https://doi.org/10.7554/eLife.70818.sa1
Author response https://doi.org/10.7554/eLife.70818.sa2

# Additional files

## Supplementary files

• Transparent reporting form

## Data availability

All data generated or analysed during this study are included in the manuscript and supporting files. Source data files have been provided for Figures 3 and 4.

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

## Appendix 1

**Appendix 1—key resources table**

| Reagent type (species) or resource | Designation | Source or reference | Identifiers | Additional information |
|---|---|---|---|---|
| strain, strain background (*Mouse*) | C57Bl/6 J | Nihon SLC | C57Bl/6JJmsSlc | |
| strain, strain background (*Rat*) | Sprague Dawley rat | Nihon SLC | Slc:SD | Time pregnant (pups were used at P0-P3 to harvest tissue for culture) |
| genetic reagent (mouse) | *Grin1*$^{flx/flx}$ | McHugh Lab (RIKEN CBS) | B6.129S4-*Grin1*$^{tm2Stl}$/J Jackson Labs 005246 | (Tsien et al. Cell 87, 1327–1338, 1996) |
| genetic reagent (mouse) | Ai9 | Itohara Lab (RIKEN CBS) | B6.Cg-*Gt(ROSA)26Sor*$^{tm9(CAG-tdTomato)Hze}$/J Jackson Labs 007909 | (Madisen et al. Nat Neurosci 13, 133–140, 2010) |
| genetic reagent (mouse) | *Grin2c*$^{iCre/+}$ | Sakimura Lab (Niigata U) | NA | (Miyazaki et al., J Neurosci 32, 1311–1328, 2012) |
| cell line (*Homo-sapiens*) | 293FT (derived from embryonic kidney) | ThermoFisher Scientific | CAT# R70007 | Used in AAV production; authenticated by STR anaylsis using GenePrint10 System (Promega) (Further details will be provided by the lead contact upon request.) |
| antibody | Anti-RFP (Chiken polyclonal) | Rockland | CAT# 600-901-379 | (IF 1:1000) |
| antibody | Anti-GFAP (Mouse monoclonal) | Synaptic Systems | CAT# 173011 RRID: AB_2232308 | (IF 1:1000) |
| antibody | Anti-GFAP (Rabbit polyoclonal) | Abcam | CAT# ab48050 RRID: AB_941765 | (IF 1:500) |
| antibody | Anti-NeuN (Recombiinant rabbit monoclonal) | Abcam | CAT# ab177487 RRID: AB_2532109 | (IF 1:500) |
| antibody | Anti-GluN1 (Guinea pig polyclonal) | Alamone Labs | CAT# AGP-046 | (IF 1:100) |
| antibody | Anti-Chicken AlexaFluor-555 (Goat polyclona) | Invitrogen | CAT# A32932 | (IF 1:1000) |
| antibody | Anti-Mouse AlexaFluor 633 (Goat polyclona) | Invitrogen | CAT# A21052 RRID: AB_2535719 | (IF 1:1000) |
| antibody | Anti-Rabbit AlexaFluor 488 (Goat polyclona) | Invitrogen | CAT# A11034 | (IF 1:1000) |
| antibody | Anti-Guinea pig AlexaFluor 647 (Goat polyclonal) | Invitrogen | CAT# A21450 RRID: AB_141882 | (IF 1:200) |
| antibody | Anti-GluN1 (Mouse monoclonal) | Synaptic Systems | CAT# 114011 RRID: AB_887750 | (WB 1:1000) (IP 5 µg / 40 µl ProteinA/G resin) |
| antibody | Anti-GluN2A/B (Rabbit polyclonal) | Synaptic Systems | CAT# 244003 RRID: AB_10804284 | (WB 1:1000) (IP 5 µg / 40 µl ProteinA/G resin) |

*Appendix 1 Continued on next page*

*Appendix 1 Continued*

| Reagent type (species) or resource | Designation | Source or reference | Identifiers | Additional information |
|---|---|---|---|---|
| antibody | Anti-GluN2C (Rabbit polyclonal) | Frontier Institute Co. | CAT# GLURE3C-RB-AF270 RRID: AB_2571763 | (WB 1:200) (IP 5 µg / 40 µl ProteinA/G resin) |
| antibody | Anti-Cre, clone 2D8 (Mouse monoclonal) | Merck Millipore | mouse anti-Cre (Merck Millipore clone 2D8) | (IP 5 µg / 40 µl ProteinA/G resin) |
| Biological sample (AAV) | AAV9.2 GFAP104-nls-mCherry-Cre | This paper | NA | Further information will be provided by the Lead Contact upon request. |
| Biological sample (AAV) | AAV9.2-GFAP104-nls-mCherry | This paper | NA | Further information will be provided by the Lead Contact upon request. |
| Biological sample (AAV) | AAVDJ8-GFAP104-nls-mCherry-Cre | Goda Lab | NA | (Letellier et al., PNAS 2016) |
| Biological sample (AAV) | AAVDJ8-GFAP104-eGFP | Goda Lab | NA | (Letellier et al., PNAS 2016) |
| sequence-based reagent | *Grin1* fw | This paper | PCR primers | |
| sequence-based reagent | *Grin1* rv | This paper | PCR primers | |
| sequence-based reagent | *Grin2a* fw | This paper | PCR primers | |
| sequence-based reagent | *Grin2a* rv | This paper | PCR primers | TTGCCCAGCTTT TCCCATTCC |
| sequence-based reagent | *Grin2b* fw | This paper | PCR primers | GGCATGATTGGT AGCTGGTC |
| sequence-based reagent | *Grin2b* rv | This paper | PCR primers | GGCTCTAAGAAG GCAGAAGGTG |
| sequence-based reagent | *Grin2c* fw | This paper | PCR primers | GGAGGCTTTCTA CAGGCATCTG |
| sequence-based reagent | *Grin2c* rv | This paper | PCR primers | ATACTTCATGTA CAGGACCCCATG |
| sequence-based reagent | *Grin2d* fw | This paper | PCR primers | |
| sequence-based reagent | *Grin2d* rv | This paper | PCR primers | |
| sequence-based reagent | *Rn28s1* fw | This paper | PCR primers | CCTACCTACTAT CCAGCGAAACC |
| sequence-based reagent | *Rn28s1* rv | This paper | PCR primers | AGCTCAACAGGG TCTTCTTTCC |
| commercial assay or kit | Single Cell-to-CT qRT-PCR Kit | Invitrogen | CAT# 4458236 | |
| commercial assay or kit | BCA Protein Assay Kit | ThermoFisher Scientific | CAT# 23,227 | |
| commercial assay or kit | TaqMan Universal PCR Master Mix | Roche Applied Biosystems | CAT# 4304437 | |
| chemical compound, drug | AlexaFluor 488 | ThermoFisher Scientific | CAT# A10436 | |

*Appendix 1 Continued on next page*

| Reagent type (species) or resource | Designation | Source or reference | Identifiers | Additional information |
|---|---|---|---|---|
| chemical compound, drug | AlexaFluor 594 | ThermoFisher Scientific | CAT# A10438 | |
| chemical compound, drug | NMDA | Tocris Bioscience | CAT# 0114 | |
| chemical compound, drug | (+)-MK 801 maleate | Tocris Bioscience | CAT# 0924 | |
| chemical compound, drug | D-AP5 | Tocris Bioscience | CAT# 0106 | |
| chemical compound, drug | QNZ46 | Tocris Bioscience | CAT# 4,810 | |
| chemical compound, drug | R025-6981 | Tocris Bioscience | CAT# 1,594 | |
| chemical compound, drug | MPEP | Tocris Bioscience | CAT# 1,212 | |
| chemical compound, drug | Picrotoxin | Sigma Aldrich | CAT# P1675 | |
| software, algorithm | Metamorph | Molecular Devices | RRID: SCR_002368 | |
| software, algorithm | pClamp | Molecular Devices | RRID: SCR_011323 | |
| software, algorithm | ImageJ | NIH | https://imagej.nih.gov/ij/ | |
| software, algorithm | OpenView | Ziv Lab (Technion) | NA | Kindly provided by Noam Ziv for use in the analysis of FM data (Kaufman et al., PLoSOne 7:e40980, 2012) |
| other | DAPI stain | Sigma Aldrich | D9564 | (1 µg/mL) |
| other | FM1-43 Dye | Invitrogen | CAT# T35356 | (10 µM) |
| other | ADVASEP-7- | Sigma Aldrich | CAT# A3723 | (1 mM) |
| other | TRIzol Reagent | Sigma Aldrich | CAT# T9424 | |
| other | cOmplete protease inhibitors | Merck (Roche Diagnostics GmbH) | CAT# 11873580001 | |
| other | EMCCD camera | Andor Technology | iXon | |
| other | Inverted Fluorescence microscope | Olympus | IX71 | |
| other | Upright Fluorescence microscope | Olympus | IX51WI | |
| other | Confocal Laser Scanning Microscope | Olympus | FV3000 | |

