## [Editor Report]

This paper provides evidence that NMDA receptors containing the GluN2C subunit are expressed in hippocampal astrocytes and are involved in maintaining a wide distribution of presynaptic release probabilities at local excitatory synapses. Theoretical modelling suggests this wide distribution of synaptic efficacy is conducive to the expression of the synaptic plasticity. It will be interesting to see if future studies support a role for astrocytic NMDA receptors in memory storage.

---

## [Decision Letter]

**Decision letter after peer review:**

Thank you for submitting your article "Astrocyte GluN2C NMDA receptors control basal synaptic strengths of hippocampal CA1 pyramidal neurons in the stratum radiatum" for consideration by *eLife*. Your article has been reviewed by 3 peer reviewers, one of whom is a member of our Board of Reviewing Editors, and the evaluation has been overseen by Lu Chen as the Senior Editor. The other reviewers have opted to remain anonymous.

Essential revisions:

The full reviews are included below to help clarify the concerns. Each of the concerns raised should be addressed, however many of these simply require clarification, minor figure edits, or additional discussion. Key revisions are related to clarifying the physiological relevance of the primary findings:

1) Addressing reviewer 2's comment #7: "A clearer hypothesis on how a NMDAR-mediated molecular signal could both reduce and increase release probability depending on its current state, or otherwise just shift synaptic Pr randomly up or down, needs to be put forward. The authors are encouraged to find some material evidence for this type of phenomenon in the literature."

2) Each reviewer had concerns regarding the modeling as presented and are in agreement that significant revision of the modeling is required. The modeling and simulations are important for providing some physiological framework for the counterintuitive result of a change in the variance but not the mean of the presynaptic release probability. The authors should focus on reviewer 2's comment #8, and reviewer 3's concerns regarding the modeling.

*Reviewer #1:*

In the current manuscript, Chipman et al. present evidence that GluN2C-containing NMDA receptors in hippocampal astrocytes tune synaptic strengths in CA1 pyramidal neurons by helping to maintain a wide distribution of presynaptic release probabilities. This work was an extension of the lab's previous work (Letellier et al. 2016) where they first showed that astrocytic calcium and NMDA receptors regulate presynaptic release probability. Here they add genetic data supporting these earlier findings and provide a model for how a narrowing of the distribution of presynaptic release probability might affect synaptic plasticity. The go on to provide strong evidence that astrocytic NMDA receptors largely contain the GluN2C subunit and provide evidence that these GluN2C-containing receptors are involved in the effect on presynaptic release probability. The conclusions of this paper are mostly supported by the data and the combination of pharmacological and genetic approaches are a major strength of this manuscript.

There are some concerns about the selectivity of some of the manipulations allowing the possibility that neuronal NMDA receptors may be involved. Further, not much new insight is provided into the mechanisms for how astrocytes might regulate the variation of presynaptic release probability, leaving this as a more descriptive work. Also, while the modeling is interesting, it was not tested experimentally using the findings presented.

Comments for the authors:

1) The combination of pharmacological and genetic approaches to blocking astrocytic NMDA receptors and GluN2C-containing receptors is a major strength of this manuscript. However, there are a few reasons to question selectivity of these manipulations on astrocytes.

First, following Cre-mediated GluN1 deletion in astrocytes a complete loss of response to the puff of NMDA/glycine would be expected, but the response was only reduced by about half (though completely eliminated by AP5) (Figure 2—figure supplement 1) raising concerns about the effectiveness of the genetic deletion.

Second, they utilize intracellular MK801 in the patch pipette to block postsynaptic NMDA receptors. It has recently been shown (Sun et al. Neuropharmacology 2018) that intracellular MK801 incompletely inhibits NMDA receptors due to an ~30,000-fold lower affinity compared to extracellular application. Indeed, they see some remaining current in Figure 1—figure supplement 1.

Furthermore, the conclusions of this paper would be better supported by a set of experiments increasing GluN2C activity rather than just reducing – perhaps with the positive allosteric modulator CIQ (Mullasseril et al. Nature Communications, 2010), or by a GluN2C replacement approach. Indeed, the statement on lines 182-193, that "these findings support the idea NMDARs play a role in bi-directionally regulating presynaptic strengths to broaden the variability of PPR" is not fully supported.

2) The modeling is interesting, but its value is limited as it was not tested experimentally using the most selective manipulations they have identified (e.g. GluN2C blockade/deletion)

3) This work remains quite descriptive without much progress in the mechanism of astrocytic control of presynaptic release probability. The authors discuss some possible mechanism, including through astrocyte-mediated release of ATP and action at adenosine receptors, release of other putative gliotransmitters, or modulation of local K^+^ dynamics. Observing GluN2C mediated effects on any of these processes would strengthen the conclusions.

*Reviewer #2:*

The present work deals with a timely and potentially important subject, the role of astroglial NMDA receptors in regulating release efficacy at local excitatory synapses. The authors combine refined electrophysiology methods with pharmacological dissection, genetic targeting, and theoretical simulations to conclude that signals mediated by astroglial NDMARs help widen the range of release probability values among CA3-CA1 synapses in the hippocampus. Modelling suggests that such widening should be conducive to the expression of synaptic plasticity. There are several issues, which the authors may want to address.

1. Whilst changes in PPR are normally inversely correlated with changes in Pr, it does not mean that PPR represents faithfully Pr across various connections, in various conditions. Slice condition, afferent excitability, retrograde messaging or network influences post-stimulus, etc. can affect PPR without affecting Pr. This has to be at least discussed, in the context.

2. Figure 1: One has to dig into the Methods to understand how PPR values were sampled. It would help the reader to mention the sampling design (number of slices / independent inputs) in the figures or in the main text. Methods: it seems that the 'cross-paired-pulse stimulation paradigm' to gauge input independence was introduced by Scimemi et al. (2004 J Neurosci).

3. The classical 'regression to the mean' phenomenon means that the random variable once sampled close to the top of its range will tend to show smaller values upon next sampling, and vice versa. This is the origin of the regression lines shown as delta-PPR plots, throughout the paper. This type of regression depends on multiple, poorly controlled concomitants, so its use for statistical inference is highly ambiguous. The authors are encouraged to focus on direct comparisons of the PPR dispersion values using appropriate, possibly non-parametric statistical tests (since some of their histograms do not look Gaussian).

4. After the NMDAR blockade in one postsynaptic cell, all other neurons will still operate their NMDARs. Their subsequent blockade might involve changes in retrograde messaging, the dynamics of extracellular potassium, etc. This has to be addressed or at least discussed.

5. Figure 3: Direct readouts of Pr and/or PPR at excitatory synapses along both apical and basal dendrites of CA1 pyramidal cells have recently been reported (Grillo et al., Nat Neurosci 2018; Jensen et al. *eLife* 2021). It would surely make sense to either adopt or at least discuss those most recent data sets in the context of the present experiments and simulations.

6. Figure 3: LTP induction by its virtue refers to a sub-population of potentiated synapses (rather than all available synapses), which is supposed to be followed by a re-distribution of synaptic weights, to avoid runaway excitation. Whether this notion is compatible with the present simulation paradigm needs explaining.

7. A clearer hypothesis on how a NMDAR-mediated molecular signal could both reduce and increase release probability depending on its current state, or otherwise just shift synaptic Pr randomly up or down, needs to be put forward. The authors are encouraged to find some material evidence for this type of phenomenon in the literature.

8. The modeling appears unnecessarily complicated yet narrow in scope. In their model, BCM plasticity reflects STDP but with uncorrelated, Poisson-process pre and postsynaptic spiking at 20 Hz, which is hardly universal. This type of models deals with a 'fractional' neurotransmitter release as a representation of Pr, and whether this could be readily combined with explicit Pr distributions to reflect reality is not clear. Perhaps a more transparent and robust, 'threshold-type' plasticity approach, such as HFS or theta-burst LTP would be helpful. Regarding the model setting per se, another recent model (https://elifesciences.org/articles/62588) gave experimentally-probed stochastic synapses with distributed Pr that are scattered along realistic CA1 pyramid dendrites. This model might be considered as the context is pretty similar.

*Reviewer #3:*

Chipman et al. conducted a battery of experiments to demonstrate that astrocytes in the CA1 area of the mouse hippocampus express NMDA-type glutamate receptors (NMDARs) containing the subunits GluN1 and GluN2C, and that signaling through these receptors influences presynaptic neurotransmitter release probability at excitatory synapses onto CA1 pyramidal neurons in the stratum radiatum region of CA1. In acute brain slices, the authors electrically stimulated presynaptic axons and recorded intracellular postsynaptic currents to monitor changes in the mean and variance of presynaptic release probability under various conditions. To isolate the effects of NMDARs not expressed in the recorded cell, the authors applied a use-dependent blocker of NMDARs intracellularly to recorded CA1 pyramidal neurons and stimulated afferent fibers until all synaptic NMDARs were silenced. On that background, application of extracellular blockers of NMDARs decreased the variance, but not the mean, of the distribution of presynaptic release probabilities measured across stimulation pathways, cells, and animals. These effects were occluded when GluN1-containing NMDARs were genetically removed specifically from hippocampal astrocytes. The authors then used single-cell RT-PCR to show that the NMDAR subunit GluN2C is expressed at higher levels in astrocytes than pyramidal neurons. A selective antagonist of GluN2C-containing NMDARs also increased the variance of presynaptic release probability in pyramidal cells, but not when GluN1-containing NMDARs were genetically removed specifically from hippocampal astrocytes. This paper replicates, corroborates, and extends previous findings from a prior publication, Letellier et al., PNAS, 2016. The rigor of these new experiments provides important additional support for an intriguing model of glial-neuronal interactions that highlights new questions regarding possible mechanisms.

To explore the functional consequences of the observed specific effect on the variance, but not the mean, of presynaptic release probability, the authors simulated models of long-term synaptic plasticity and suggested that the variance of release probability affects changes in synaptic weight, and by implication, memory storage. This paper could, in principle, be enhanced by the inclusion of this type of modeling. However, in the current form, the modeling component is incompletely motivated, insufficiently described and interpreted, and may suffer from technical errors.

Comments for the authors:

Please find below comments, questions, constructive criticism, and suggestions to potentially improve the manuscript and enhance its clarity, interpretability, and impact for a broad readership.

Major commentary on Figure 3:

Ideally a model of synaptic transmission with use-dependent release probability would first be fit to and compared to the experimental data, and then used to explore the implications of the experimental findings. It is indeed a counter-intuitive result that signaling in hippocampal astrocytes alters the variance, but not the mean, of the distribution of presynaptic release probabilities at excitatory synapses onto CA1 pyramidal neurons, and it is not obvious how this would influence information processing. This is a great opportunity to use modeling to gain insight into how such a change would impact circuit function and/or memory storage. If done well, it could potentially greatly elevate the impact of the paper. However, in current form, the text does not sufficiently motivate the use of the specific plasticity models chosen, the presentation of the modeling data is confusing and hard to relate to the experimental results, and there appears to be a technical error that confounds the results. Please find below suggestions to improve this component of the paper.

1) In the experiments presented in Figures 1B – D, a small number of fibers are stimulated, and the ratio of responses to a 50 ms paired-pulse stimulation is measured. Each measurement reflects the compound effect of multiple synapses that themselves might have different initial release probabilities (Pr), and different degrees of facilitation or depression. Across many stimulation pathways, cells, and animals, a distribution of these paired-pulse ratios (PPR) is obtained and analyzed. These distributions appear to be fit by a normal distribution. However, in Figure 3A, a population of synapses is simulated, with their individual release probabilities sampled from a log-normal distribution. The first step should be to select random groups of these synapses with release probabilities sampled from the modeled distribution, probe them with a paired-pulse protocol, and verify that indeed a normal distribution of PPRs are obtained that recapitulate the experimental data in Figure 1B. Then, an appropriate distribution of release probabilities with a lower variance can be calibrated to simulate the AP5 condition. Again a paired-pulse protocol should be used to verify that the resulting PPR distribution is normal, has the same mean as the control condition, but a lower variance recapitulating the data shown in Figure 1D.

Currently, the way that the authors have constructed their release probability distributions, there is a parameter U_max_ that sets the value of U_SE_ where the distribution is maximal. A Control distribution with large variance (σ_max_) and a given U_max_ can be compared to an AP5 distribution with lower variance (σ_max_ / 2), but the same U_max_. However, the mean U_SE_ sampled from these two distributions with the same U_max_ are not equal. The mean U_SE_ sampled from the Control log-normal distribution is higher than the mean U_SE_ sampled from the AP5 distribution. So the authors have not succeeded in modeling the situation where AP5 changes only the variance, but not the mean Pr. As an aside, the definition of "SE" in U_SE_ is never provided in the paper.

2) The Equations 5, 6, and 7 must contain an error. I was not able to reproduce exactly the distributions shown in Figure 1A with U_max_ = 0.1. Also the value of the "a" term in Equation 6 is not provided.

3) Currently, the choice to test the BCM plasticity model is not motivated at all in the text. On the surface, this is an odd choice of model to test because it has both a Hebbian component that depends on coincidence of pre- and post-synaptic activity, but also a homeostatic, meta-plastic component that adjusts the transition point from potentiation to depression depending on the recent history of post-synaptic activation. While most implementations of the BCM model assume an effective presynaptic release probability of 1, here the authors would like to investigate how a dynamic release probability affects the changes in synaptic weight produced by this plasticity model. So the first test should be to compare what happens when the stimulated synapses with the same initial weight have a low U_SE_ vs. a high U_SE_. Traces for all relevant model intermediates over time should be shown: presynaptic rate r(t), successful release event rate x(t), post-synaptic current I(t), dynamic plasticity threshold Theta(t), and the synaptic weight w(t). My intuition is that synapses with a lower U_SE_ will result in a lower mean x(t), resulting in a lower mean I(t), and initially favor synaptic depression. However, over time, this will cause Theta(t) to decrease, causing potentiation to be preferred over depression. It is possible that, given enough time to equilibrate, the final target weight will be identical for low U_SE_ vs. high U_SE_, as long as the BCM rule is given enough time to equilibrate.

Once it has been established how the BCM rule is sensitive to U_SE_, then the behavior of the learning rule can be probed with a random group of synapses with variable U_SE_ sampled from an appropriate distribution. However, I suspect that as long as the mean U_SE_ is truly equal between two distributions, then the average equilibrium weight across the sample of synapses will be equal, regardless of the variance of the U distribution. The changes at low and high U_SE_ will cancel each other out. It is critical that the authors ensure that they are comparing distributions with equal mean U_SE_. If there truly is an effect of the variance of the distribution separately from the mean, then the authors are advised to provide substantially more interpretation and discussion of how or why this is the case. It is not sufficient to simply show the modeling results and state that variance matters without explaining the mechanism.

4) In Figure 3B, the value of alpha is varied, but this parameter is not sufficiently described in the text, figure legend, or Methods. This should not be used as a parameter to shift the balance between potentiation and depression. Instead, it should be the Pr (U_SE_) that determines x(t) and I(t), and then it is the relative value of I(t) and Theta(t) determines the balance between potentiation and depression. An alternative could be to compare different initial values of Theta(t), though this will quickly equilibrate to differences in the mean I(t) that result from differences in U_SE_.

5) In Figure 3B, the sigma symbol with a subscript of 0.1 is used in the figure label but not anywhere in the text, figure legend, or Methods. I believe this is meant to indicate the value of σ_max_ when U_max_ = 0.1. This should be made explicit in the text.

6) In Figure 3C, what variable was changed to generate different values for the absolute changes in synaptic weight in the Control condition (the x-axis)? Was it the value of "alpha"? This was not explained sufficiently. This plot does not have an obvious interpretation, and one is not provided in the text. It does not help the reader understand how the variance of the U_SE_ distribution would cause this particular change in long-term plasticity.

7) With regards to the use of an STDP model in Figure 3D – again, there is not sufficient description the text to understand how this model is implemented. Why is the x-axis labeled "Δt / ms"? The use of the variable postsynaptic current A_2_, or why the y-axis is labeled "A_2_ / ms" is not clear at all.

A classical phenomenological STDP model would stimulate presynaptic spikes at defined times relative to postsynaptic spikes, and update the synaptic weight(s) accordingly. Inclusion of a dynamic release probability would simply cause some fraction of the presynaptic spikes to be failures and therefore to not be paired with a postsynaptic spike within the plasticity window, and therefore not result in any change in weight. Again my intuition is that for a synapse with a low Pr (U_SE_), it will take more spike pairings over a longer time to equilibrate compared to a synapse with a high Pr. So the first test should be to compare the final weights of synapses with low U_SE_ to those of synapses with high U_SE_ after a period of Poisson pre- and post-synaptic spiking. However, it is again not clear that changes in the variance of the U_SE_ distribution in the absence of any changes in the mean would have any impact on the relative degree of potentiation or depression.

[Editors' note: further revisions were suggested prior to acceptance, as described below.]

Thank you for resubmitting your work entitled "Astrocyte GluN2C NMDA receptors control basal synaptic strengths of hippocampal CA1 pyramidal neurons in the stratum radiatum" for further consideration by *eLife*. Your revised article has been evaluated by Lu Chen as the Senior Editor and John Gray as the Reviewing Editor in consultation with the original peer reviewers.

The authors have made a significant effort in addressing most of the concerned expressed by the reviewers and the manuscript has been improved, but there are some remaining issues that need to be addressed, as outlined below:

1) At the suggestion of the reviewers, the authors have revised their modeling of the potential impact of changing Pr distribution (but not mean) on synaptic plasticity. Appropriately, they removed the BCM modeling and focused on STDP and provided experimentally collected estimates of Pr using FM1-43 imaging of the RRP from hippocampal neurons co-cultured with astrocytes. However, the reviewers found the text related to Figure 3 to be confusing and poorly described. Specifically, it took some discussion for all the reviewers to understand how Figure 3 actually tests the impact of Pr distribution based on the A_2_ injection amplitude. That is, if you have two distributions of Pr with the same mean, but one has a wide variance and one has a narrow variance – if you set a threshold above the mean, the probability that the threshold will be crossed is higher for the distribution with wider variance. This needs to be explicitly described and explained by the authors in the manuscript text and in the figure legend. Figure revisions are not necessary, but could also benefit the readers if there is improved clarity.

2) The authors are also urged to mention explicitly that their delta PPR graphs contain a regression-to-the-mean component which is however effectively cancelled out when making direct graph comparisons, within the same parametric space

---

## [Author Response]

Reviewer #1:[…] There are some concerns about the selectivity of some of the manipulations allowing the possibility that neuronal NMDA receptors may be involved. Further, not much new insight is provided into the mechanisms for how astrocytes might regulate the variation of presynaptic release probability, leaving this as a more descriptive work. Also, while the modeling is interesting, it was not tested experimentally using the findings presented.Comments for the authors:1) The combination of pharmacological and genetic approaches to blocking astrocytic NMDA receptors and GluN2C-containing receptors is a major strength of this manuscript. However, there are a few reasons to question selectivity of these manipulations on astrocytes.First, following Cre-mediated GluN1 deletion in astrocytes a complete loss of response to the puff of NMDA/glycine would be expected, but the response was only reduced by about half (though completely eliminated by AP5) (Figure 2—figure supplement 1) raising concerns about the effectiveness of the genetic deletion.

The NMDA/glycine puff is expected to activate nearby neuronal NMDARs that are not targeted by the Cre-mediated knockdown. Postsynaptic NMDA receptor activation has been shown to be the major source of K^+^ efflux (Shin et al., Cell Rep 2013), which will alter the local extracellular ionic milieu, and in turn, impact astrocyte ionic currents. In our prior work (Letellier et al., 2016), we demonstrated that the slow depolarization of astrocyte membrane potential induced by NMDA/glycine puff could be partially blocked by extracellular application of nifedipine or nimodipine, and intracellular loading into astrocytes of QX-314 or D-890, an L-VGCC antagonist. Therefore, other voltage-gated currents, including calcium channels contribute to the NMDA/glycine puff responses in astrocytes. We have clarified the sentence in lines 223-227:

“The efficacy of GluN1 knock-down in astrocytes was assessed electrophysiologically by patch-clamping astrocytes and monitoring slow depolarizing responses elicited by puff applying NMDA and glycine (1mM each) (Figure 2—figure supplement 1) which were mediated by NMDARs but were also contributed in part by voltage-gated calcium channels in astrocytes (Letellier et al., 2016)”.

Please note that we have included the NMDA/glycine puff response data in the present manuscript to demonstrate the ability to detect AP5-dependent responses across astrocytes in all three layers in the CA1 subfield, which is in line with the RT-PCR results. Importantly, the data highlight the differences in sensitivity of the NMDA/glycine puff responses to astrocyte NMDAR knock-down across the three layers, which suggest that signaling downstream of NMDAR activation underlying the slow depolarization differs across the three layers, specifically for astrocytes in the *stratum radiatum*.

Second, they utilize intracellular MK801 in the patch pipette to block postsynaptic NMDA receptors. It has recently been shown (Sun et al. Neuropharmacology 2018) that intracellular MK801 incompletely inhibits NMDA receptors due to an ~30,000-fold lower affinity compared to extracellular application. Indeed, they see some remaining current in Figure 1—figure supplement 1.

We acknowledge that MK-801 may not always result in 100% block of postsynaptic NMDAR currents. After confirming the paucity of expression of GluN2C in pyramidal neurons and GluN2D in hippocampal extracts, and the robust expression of GluN2C in astrocytes, we find that application of GluN2C/D-specific inhibitor is sufficient to alter PPR. This points to the involvement of astrocyte GluN2C NMDAR. Moreover, the effect on PPR is occluded by the genetic knock-down of GluN1 in astrocytes. Collectively, these observations support the sufficiency of astrocyte NMDARs in affecting PPR.

Furthermore, the conclusions of this paper would be better supported by a set of experiments increasing GluN2C activity rather than just reducing – perhaps with the positive allosteric modulator CIQ (Mullasseril et al. Nature Communications, 2010), or by a GluN2C replacement approach. Indeed, the statement on lines 182-193, that "these findings support the idea NMDARs play a role in bi-directionally regulating presynaptic strengths to broaden the variability of PPR" is not fully supported.

We agree with the reviewer that expanding the pharmacological manipulation of GluN2C activity by using CIQ to enhance its activity and genetically manipulating GluN2C expression might provide additional experimental support to the present findings. Nevertheless, gain of function experiments require careful considerations. We plan to capitalize on such tools for our future work in pursuing the mechanisms underlying GluN2C-NMDAR-dependent presynaptic modulation. We have toned down the statement on lines 204-206 to read:

“Collectively, these findings support the idea that NMDARs play a role in regulating presynaptic strengths to broaden the variability of PPR without appreciably impacting the mean PPR”.

2) The modeling is interesting, but its value is limited as it was not tested experimentally using the most selective manipulations they have identified (e.g. GluN2C blockade/deletion)3) This work remains quite descriptive without much progress in the mechanism of astrocytic control of presynaptic release probability. The authors discuss some possible mechanism, including through astrocyte-mediated release of ATP and action at adenosine receptors, release of other putative gliotransmitters, or modulation of local K^+^ dynamics. Observing GluN2C mediated effects on any of these processes would strengthen the conclusions.

We respectfully disagree with the reviewer that the modeling is of limited value. The modeling provides a framework to demonstrate the impact of directly altering the width of release probability variance on synaptic plasticity. The experimental test of GluN2C manipulation on synaptic plasticity involving careful characterization of the underlying mechanisms, including the determination of glial signaling and putative gliotransmitters, are aspects of the study that we will pursue in the future and are beyond the scope of the present work. Please note that we have made major changes to the modelling section in the revised manuscript as described in detail in the replies to Reviewer #3.

Reviewer #2:The present work deals with a timely and potentially important subject, the role of astroglial NMDA receptors in regulating release efficacy at local excitatory synapses. The authors combine refined electrophysiology methods with pharmacological dissection, genetic targeting, and theoretical simulations to conclude that signals mediated by astroglial NDMARs help widen the range of release probability values among CA3-CA1 synapses in the hippocampus. Modelling suggests that such widening should be conducive to the expression of synaptic plasticity. There are several issues, which the authors may want to address.1. Whilst changes in PPR are normally inversely correlated with changes in Pr, it does not mean that PPR represents faithfully Pr across various connections, in various conditions. Slice condition, afferent excitability, retrograde messaging or network influences post-stimulus, etc. can affect PPR without affecting Pr. This has to be at least discussed, in the context.We fully acknowledge the fact that PPR may not necessarily always relate to Pr. We have amended the discussion as follows in lines 484-491:

“The present study took advantage of paired-pulse response as a measure of presynaptic efficacy. […] Therefore, in some experiments, presumed changes in release probability as monitored by PPR were further corroborated by the correlated changes in CV^-2^ (Malinow and Tsien, 1990; Larkman et al., 1992).”

2. Figure 1: One has to dig into the Methods to understand how PPR values were sampled. It would help the reader to mention the sampling design (number of slices / independent inputs) in the figures or in the main text. Methods: it seems that the 'cross-paired-pulse stimulation paradigm' to gauge input independence was introduced by Scimemi et al. (2004 J Neurosci).

Number of inputs, cells and mice are stated in the bottom of figure legend. Cross-paired pulse stimulation has been used as a standard test for input independence for quite some time (e.g. Otani and Connor, J Physiol 1996). We now cite Otani and Connor (1996) and Scimemi et al. (2004) as example references in line 669.

3. The classical 'regression to the mean' phenomenon means that the random variable once sampled close to the top of its range will tend to show smaller values upon next sampling, and vice versa. This is the origin of the regression lines shown as delta-PPR plots, throughout the paper. This type of regression depends on multiple, poorly controlled concomitants, so its use for statistical inference is highly ambiguous. The authors are encouraged to focus on direct comparisons of the PPR dispersion values using appropriate, possibly non-parametric statistical tests (since some of their histograms do not look Gaussian).

We thank the reviewer for the important point. We included the delta-PPR data in order to consider the present results relative to data in our previous study that used delta-PPR comparisons (Letellier et al., 2016). In the present work we sought to compare PPR distributions directly as such an analysis could be more informative, as the reviewer indicated. The datasets were tested for normality using the Shapiro-Wilk test. Details of statistical analyses are described in Materials and methods section, and the statistics of individual datasets are included in the excel file labelled “E-phys statistics report”.

4. After the NMDAR blockade in one postsynaptic cell, all other neurons will still operate their NMDARs. Their subsequent blockade might involve changes in retrograde messaging, the dynamics of extracellular potassium, etc. This has to be addressed or at least discussed.

We acknowledge the fact that we cannot exclude the possible contribution of NMDARs present in neurons that we are not recording from nor of NMDARs in the recorded neuron should there be incomplete block of NMDARs from intracellularly applied MK801. Please see the reply to reviewer 1, second comment of part 1.

5. Figure 3: Direct readouts of Pr and/or PPR at excitatory synapses along both apical and basal dendrites of CA1 pyramidal cells have recently been reported (Grillo et al., Nat Neurosci 2018; Jensen et al. eLife 2021). It would surely make sense to either adopt or at least discuss those most recent data sets in the context of the present experiments and simulations.

For the mathematical modelling in the revised manuscript, we experimentally obtained the Pr distribution. Our Pr estimate was based on direct measurements using the FM1-43 styryl dye, of readily releasable vesicle pool (RRP) at individual boutons formed on dendrites of hippocampal pyramidal neurons in primary culture. FM dye experiments have been carried out following standard procedures in the presence of 10 μM CNQX to block AMPA receptors so as to prevent recurrent network activity. Under such conditions, neuronal NMDARs do not experience the efficient removal of Mg block which is facilitated by membrane depolarization from AMPAR activation. In contrast, GluN2C-NMDARs have reduced Mg block compared to neuronally expressed NMDARs, and given the expression of GluN2C in hippocampal astrocytes, any effect of AP5 (added on top of CNQX), if observed, could be associated with astrocyte NMDAR inhibition. Notably, the distribution of RRP monitored in the presence of AP5 and CNQX was less variable compared to the distribution observed in CNQX alone.

With respect to Grillo et al. (2018) and Jensen et al. (2021) studies, we refer to the work in the discussion (lines 503-506):

“The use of methods that allow for a more direct estimate of release probability in hippocampal CA1 neurons as have been reported recently (e.g. Grillo et al., 2018; Jensen et al., 2021) will be informative in further pursuing the underlying mechanisms.”

6. Figure 3: LTP induction by its virtue refers to a sub-population of potentiated synapses (rather than all available synapses), which is supposed to be followed by a re-distribution of synaptic weights, to avoid runaway excitation. Whether this notion is compatible with the present simulation paradigm needs explaining.

We have not considered homeostatic compensation across space in the present simulation. Please note that in the Poisson spike model where presynaptic and postsynaptic neurons are not fixed to spike in any particular order or frequency to favor LTP or LTD induction, both forms of plasticity are observed across trials. Thus potentiation could be countered by depression and vice versa across time at minimum. Importantly, the extent expression of both LTP and LTD are compromised by reducing the variability of presynaptic release probability.

7. A clearer hypothesis on how a NMDAR-mediated molecular signal could both reduce and increase release probability depending on its current state, or otherwise just shift synaptic Pr randomly up or down, needs to be put forward. The authors are encouraged to find some material evidence for this type of phenomenon in the literature.

We apologize for the lack of clarity in our original discussion of how astrocyte NMDAR activity could expand the range of release probability. The relevant section (lines 533-560) has been reworded with an additional reference (Darabid et al., Cell Rep 2018):

“The mechanism by which astrocyte GluN2C NMDARs are coupled to the changes in presynaptic efficacy remains to be clarified. […] Further understanding of such mechanisms is warranted to identify the basis for how the relative differences in synaptic strengths are promoted to facilitate broadening of the range of presynaptic efficacy.”

8. The modeling appears unnecessarily complicated yet narrow in scope. In their model, BCM plasticity reflects STDP but with uncorrelated, Poisson-process pre and postsynaptic spiking at 20 Hz, which is hardly universal. This type of models deals with a 'fractional' neurotransmitter release as a representation of Pr, and whether this could be readily combined with explicit Pr distributions to reflect reality is not clear. Perhaps a more transparent and robust, 'threshold-type' plasticity approach, such as HFS or theta-burst LTP would be helpful. Regarding the model setting per se, another recent model (https://elifesciences.org/articles/62588) gave experimentally-probed stochastic synapses with distributed Pr that are scattered along realistic CA1 pyramid dendrites. This model might be considered as the context is pretty similar.

We have decided to forego the BCM model in the present study given the concerns raised by the reviewers, and instead, we have sought to simplify the modelling by focusing on leaky integrate-and-fire neurons under two different conditions: first, by applying the commonly used STDP paradigm, and second, for exploring a more general condition by applying Poisson spike trains with an average spike rate of 10 Hz. Moreover, we took advantage of an experimentally determined Pr distributions which were obtained under the control condition and in the presence of AP5 that yielded less variable Pr as explained above. In the improved version of the STDP modeling study, we found a clear picture that the reduction of variance in release probability can suppress the efficacy of long-term potentiation and depression.

Reviewer #3:Chipman et al. conducted a battery of experiments to demonstrate that astrocytes in the CA1 area of the mouse hippocampus express NMDA-type glutamate receptors (NMDARs) containing the subunits GluN1 and GluN2C, and that signaling through these receptors influences presynaptic neurotransmitter release probability at excitatory synapses onto CA1 pyramidal neurons in the stratum radiatum region of CA1. In acute brain slices, the authors electrically stimulated presynaptic axons and recorded intracellular postsynaptic currents to monitor changes in the mean and variance of presynaptic release probability under various conditions. To isolate the effects of NMDARs not expressed in the recorded cell, the authors applied a use-dependent blocker of NMDARs intracellularly to recorded CA1 pyramidal neurons and stimulated afferent fibers until all synaptic NMDARs were silenced. On that background, application of extracellular blockers of NMDARs decreased the variance, but not the mean, of the distribution of presynaptic release probabilities measured across stimulation pathways, cells, and animals. These effects were occluded when GluN1-containing NMDARs were genetically removed specifically from hippocampal astrocytes. The authors then used single-cell RT-PCR to show that the NMDAR subunit GluN2C is expressed at higher levels in astrocytes than pyramidal neurons. A selective antagonist of GluN2C-containing NMDARs also increased the variance of presynaptic release probability in pyramidal cells, but not when GluN1-containing NMDARs were genetically removed specifically from hippocampal astrocytes. This paper replicates, corroborates, and extends previous findings from a prior publication, Letellier et al., PNAS, 2016. The rigor of these new experiments provides important additional support for an intriguing model of glial-neuronal interactions that highlights new questions regarding possible mechanisms.To explore the functional consequences of the observed specific effect on the variance, but not the mean, of presynaptic release probability, the authors simulated models of long-term synaptic plasticity and suggested that the variance of release probability affects changes in synaptic weight, and by implication, memory storage. This paper could, in principle, be enhanced by the inclusion of this type of modeling. However, in the current form, the modeling component is incompletely motivated, insufficiently described and interpreted, and may suffer from technical errors.

We thank the reviewer for acknowledging that our new experiments extend our previous work and provide “important additional support for an intriguing model of glial-neuronal interactions”. The reviewer has also indicated several major concerns for the results presented in Figure 3 of the original submission. Following his/her suggestions, we have substantially revised the modelling part of the study.

Comments for the authors:Please find below comments, questions, constructive criticism, and suggestions to potentially improve the manuscript and enhance its clarity, interpretability, and impact for a broad readership.Major commentary on Figure 3:Ideally a model of synaptic transmission with use-dependent release probability would first be fit to and compared to the experimental data, and then used to explore the implications of the experimental findings. It is indeed a counter-intuitive result that signaling in hippocampal astrocytes alters the variance, but not the mean, of the distribution of presynaptic release probabilities at excitatory synapses onto CA1 pyramidal neurons, and it is not obvious how this would influence information processing. This is a great opportunity to use modeling to gain insight into how such a change would impact circuit function and/or memory storage. If done well, it could potentially greatly elevate the impact of the paper. However, in current form, the text does not sufficiently motivate the use of the specific plasticity models chosen, the presentation of the modeling data is confusing and hard to relate to the experimental results, and there appears to be a technical error that confounds the results. Please find below suggestions to improve this component of the paper.1) In the experiments presented in Figures 1B – D, a small number of fibers are stimulated, and the ratio of responses to a 50 ms paired-pulse stimulation is measured. Each measurement reflects the compound effect of multiple synapses that themselves might have different initial release probabilities (Pr), and different degrees of facilitation or depression. Across many stimulation pathways, cells, and animals, a distribution of these paired-pulse ratios (PPR) is obtained and analyzed. These distributions appear to be fit by a normal distribution. However, in Figure 3A, a population of synapses is simulated, with their individual release probabilities sampled from a log-normal distribution. The first step should be to select random groups of these synapses with release probabilities sampled from the modeled distribution, probe them with a paired-pulse protocol, and verify that indeed a normal distribution of PPRs are obtained that recapitulate the experimental data in Figure 1B. Then, an appropriate distribution of release probabilities with a lower variance can be calibrated to simulate the AP5 condition. Again a paired-pulse protocol should be used to verify that the resulting PPR distribution is normal, has the same mean as the control condition, but a lower variance recapitulating the data shown in Figure 1D.Currently, the way that the authors have constructed their release probability distributions, there is a parameter U_max_ that sets the value of U_SE_ where the distribution is maximal. A Control distribution with large variance (σ_max_) and a given U_max_ can be compared to an AP5 distribution with lower variance (σ_max_ / 2), but the same U_max_. However, the mean U_SE_ sampled from these two distributions with the same U_max_ are not equal. The mean U_SE_ sampled from the Control log-normal distribution is higher than the mean U_SE_ sampled from the AP5 distribution. So the authors have not succeeded in modeling the situation where AP5 changes only the variance, but not the mean Pr. As an aside, the definition of "SE" in U_SE_ is never provided in the paper.

We thank the reviewer for the criticisms and helpful suggestions. We have completely revamped our approach in the numerical modeling section. One of the major changes is in the way we obtained the Pr distribution, which was based on our own experimental measurements that used a styryl dye FM1-43 to label the readily releasable pool at individual boutons in cultured hippocampal pyramidal neurons. The readily releasable pool provides a more direct estimate Pr compared to PPR. As an advantage of this approach, we could obtain the Pr distributions experimentally under both control condition and in the presence of AP5, and particularly in AP5, the distribution showed a reduced variance (Figure 3—figure supplement 1). This result was consistent with acute slice experiments presented. Given that FM1-43 experiments were performed under conditions that did not favor the activation of neuronal NMDARs (i.e. in the continuous presence of CNQX), the effect of AP5 suggested the consequences of blocking astrocyte NMDARs. Furthermore, in our prior study, co-cultures of hippocampal neurons and astrocytes preserved aspects of astrocyte NMDAR signaling-dependent regulation of PPR (Letellier et al., 2016). Therefore, collectively, the FM1-43 data support the main conclusions and provide experimental Pr distributions for both the control and the AP5 cases. In the revised manuscript, we obtained gamma distribution fits to the FM1-43 experimental data, which were then used to perform numerical simulations. Following the suggestions, the simulations have been tested for the fits to experimentally obtained EPSC peaks, and simulated PPR has been compared to data reported in Figure 1B-D in control and AP5 (Figure 3—figure supplement 2).

2) The Equations 5, 6, and 7 must contain an error. I was not able to reproduce exactly the distributions shown in Figure 1A with U_max_ = 0.1. Also the value of the "a" term in Equation 6 is not provided.

In the revised manuscript, we have used the gamma distributions for simulations, and therefore the equations referred to are not included. The parameters for the gamma distributions are obtained from fitting the FM1-43 signal as explained above, and the values are stated in the legend of Figure 3—figure supplement 1 and in Materials and methods.

3) Currently, the choice to test the BCM plasticity model is not motivated at all in the text. On the surface, this is an odd choice of model to test because it has both a Hebbian component that depends on coincidence of pre- and post-synaptic activity, but also a homeostatic, meta-plastic component that adjusts the transition point from potentiation to depression depending on the recent history of post-synaptic activation. While most implementations of the BCM model assume an effective presynaptic release probability of 1, here the authors would like to investigate how a dynamic release probability affects the changes in synaptic weight produced by this plasticity model. So the first test should be to compare what happens when the stimulated synapses with the same initial weight have a low U_SE_ vs. a high U_SE_. Traces for all relevant model intermediates over time should be shown: presynaptic rate r(t), successful release event rate x(t), post-synaptic current I(t), dynamic plasticity threshold Theta(t), and the synaptic weight w(t). My intuition is that synapses with a lower U_SE_ will result in a lower mean x(t), resulting in a lower mean I(t), and initially favor synaptic depression. However, over time, this will cause Theta(t) to decrease, causing potentiation to be preferred over depression. It is possible that, given enough time to equilibrate, the final target weight will be identical for low U_SE_ vs. high U_SE_, as long as the BCM rule is given enough time to equilibrate.Once it has been established how the BCM rule is sensitive to U_SE_, then the behavior of the learning rule can be probed with a random group of synapses with variable U_SE_ sampled from an appropriate distribution. However, I suspect that as long as the mean U_SE_ is truly equal between two distributions, then the average equilibrium weight across the sample of synapses will be equal, regardless of the variance of the U distribution. The changes at low and high U_SE_ will cancel each other out. It is critical that the authors ensure that they are comparing distributions with equal mean U_SE_. If there truly is an effect of the variance of the distribution separately from the mean, then the authors are advised to provide substantially more interpretation and discussion of how or why this is the case. It is not sufficient to simply show the modeling results and state that variance matters without explaining the mechanism.

We acknowledge the issues concerning the BCM model. To present a clearer simulation study we removed the BCM model in the revised manuscript. We focused on the model of spiking neurons with STDP to demonstrate how the range of release probability variability could affect the efficacy of STDP.

4) In Figure 3B, the value of alpha is varied, but this parameter is not sufficiently described in the text, figure legend, or Methods. This should not be used as a parameter to shift the balance between potentiation and depression. Instead, it should be the Pr (U_SE_) that determines x(t) and I(t), and then it is the relative value of I(t) and Theta(t) determines the balance between potentiation and depression. An alternative could be to compare different initial values of Theta(t), though this will quickly equilibrate to differences in the mean I(t) that result from differences in U_SE_.

As the BCM model is removed, original Figure 3B is no longer shown. However, to address the comments in general, we have included more descriptions of the model. Also, as explained above, we have included the FM1-43 data and the corresponding fits to justify the parameters of the gamma distributions used in simulations.

5) In Figure 3B, the sigma symbol with a subscript of 0.1 is used in the figure label but not anywhere in the text, figure legend, or Methods. I believe this is meant to indicate the value of σ_max_ when U_max_ = 0.1. This should be made explicit in the text.

Original Figure 3B has been deleted.

6) In Figure 3C, what variable was changed to generate different values for the absolute changes in synaptic weight in the Control condition (the x-axis)? Was it the value of "alpha"? This was not explained sufficiently. This plot does not have an obvious interpretation, and one is not provided in the text. It does not help the reader understand how the variance of the U_SE_ distribution would cause this particular change in long-term plasticity.

Original Figure 3C has been deleted.

7) With regards to the use of an STDP model in Figure 3D – again, there is not sufficient description the text to understand how this model is implemented. Why is the x-axis labeled "Δt / ms"? The use of the variable postsynaptic current A_2_, or why the y-axis is labeled "A_2_ / ms" is not clear at all.A classical phenomenological STDP model would stimulate presynaptic spikes at defined times relative to postsynaptic spikes, and update the synaptic weight(s) accordingly. Inclusion of a dynamic release probability would simply cause some fraction of the presynaptic spikes to be failures and therefore to not be paired with a postsynaptic spike within the plasticity window, and therefore not result in any change in weight. Again my intuition is that for a synapse with a low Pr (U_SE_), it will take more spike pairings over a longer time to equilibrate compared to a synapse with a high Pr. So the first test should be to compare the final weights of synapses with low U_SE_ to those of synapses with high U_SE_ after a period of Poisson pre- and post-synaptic spiking. However, it is again not clear that changes in the variance of the U_SE_ distribution in the absence of any changes in the mean would have any impact on the relative degree of potentiation or depression.

Figure 3D in the original submission is now presented as Figure 3B in the revision. We apologize for the typo in the vertical axis of the previous version, which has now been corrected. This plot shows the resulting synaptic weight after spike pairing, with different postsynaptic current injection (A_2_) and spike-pairing time difference (Δt) with the aim to demonstrate how the change in Pr distribution can impact the sensitivity of STDP expression. As pointed out by the reviewer, we acknowledge that there could be some chance that the presynaptic spike has no effect due to the small initial release probability. Here we present the mean of 10,000 simulations, and the result shows the differences in the general tendency of undergoing potentiation associated with Pr distributions representing three different conditions: control, AP5, and AP5 condition in which the mean is constrained to the mean of the control Pr distribution. The third condition has been implemented to address the concerns related to the contribution of the mean of distribution. We have made extensive changes to the text to clarify the model implementations shown in Figures 3 and 4 with added descriptions. Specifically with respect to Figure 3B, in order to highlight the difference between three conditions shown, we now include a bar plot (new Figure 3C) which shows the average maximum change of synaptic weight when the extra input to the post-synaptic neuron is small. We feel that these changes have substantially clarified the key consequences of altering the Pr variance on the expression of plasticity.

[Editors' note: further revisions were suggested prior to acceptance, as described below.]

The authors have made a significant effort in addressing most of the concerned expressed by the reviewers and the manuscript has been improved, but there are some remaining issues that need to be addressed, as outlined below:1) At the suggestion of the reviewers, the authors have revised their modeling of the potential impact of changing Pr distribution (but not mean) on synaptic plasticity. Appropriately, they removed the BCM modeling and focused on STDP and provided experimentally collected estimates of Pr using FM1-43 imaging of the RRP from hippocampal neurons co-cultured with astrocytes. However, the reviewers found the text related to Figure 3 to be confusing and poorly described. Specifically, it took some discussion for all the reviewers to understand how Figure 3 actually tests the impact of Pr distribution based on the A2 injection amplitude. That is, if you have two distributions of Pr with the same mean, but one has a wide variance and one has a narrow variance – if you set a threshold above the mean, the probability that the threshold will be crossed is higher for the distribution with wider variance. This needs to be explicitly described and explained by the authors in the manuscript text and in the figure legend. Figure revisions are not necessary, but could also benefit the readers if there is improved clarity.

Additional explanations to clarify Figure 3B have been added in the main text and in the legend. Moreover, color coding used in Figure 3B has been adjusted to improve clarity. Please note that a finer color gradation scheme allowed for visualization of small synaptic weight changes that were not previously visible. For this reason, relative to the previous Figure 3B, there are very small shifts in the position of the arrows, although the shifts do not affect the conclusions.

2) The authors are also urged to mention explicitly that their delta PPR graphs contain a regression-to-the-mean component which is however effectively cancelled out when making direct graph comparisons, within the same parametric space

The statement regarding delta PPR plots has been added to the Results section when we first introduce the delta PPR plots.